# Red and blue language: Word choices in the Trump and Harris 2024 presidential debate

**Philipp Wicke**[1,2☯], **Marianna M. Bolognesi**[3☯*]

1 Institute for Information and Language Processing (CIS), LMU Munich, Munich, Bavaria, Germany,
2 Munich Center for Machine Learning (MCML), Munich, Bavaria, Germany, 3 Department of Modern
Languages, Literatures and Cultures, University Bologna, Bologna, Italy

☯ These authors contributed equally to this work.
* m.bolognesi@unibo.it

and blue language: Word choices in the Trump
and Harris 2024 presidential debate. PLoS One
20(6): e0324715.

Technology, JAPAN

**Peer Review History:** PLOS recognizes the
benefits of transparency in the peer review
process; therefore, we enable the publication of
all of the content of peer review and author
responses alongside final, published articles.
The editorial history of this article is available
here: https://doi.org/10.1371/journal.pone.
0324715

**Data availability statement:** All of our code is
publicly available at: https:
//github.com/PhilWicke/debate_analysis24. All

## Abstract

Political debates are a peculiar type of political discourse, in which candidates directly
confront one another, addressing not only the the moderator's questions, but also their
opponent's statements, as well as the concerns of voters from both parties and unde-
cided voters. Therefore, language is adjusted to meet specific expectations and achieve
persuasion. We analyse how the language of Trump and Harris during the Presidential
debate (September 10th, 2024) differs in relation to semantic and pragmatic features,
for which we formulated targeted hypotheses: framing values and ideology, appealing
to emotion, using words with different degrees of concreteness and specificity, address-
ing others through singular or plural pronouns. Our findings include: differences in the
use of figurative frames (Harris often framing issues around recovery and empower-
ment, Trump often focused on crisis and decline); similar use of emotional language, with
Trump showing a slightly higher tendency toward negativity and toward less subjective
language compared to Harris; no significant difference in the specificity of candidates'
responses; similar use of abstract language, with Trump showing more variability than
Harris, depending on the subject discussed; differences in addressing the opponent, with
Trump not mentioning Harris by name, while Harris referring to Trump frequently; differ-
ent uses of pronouns, with Harris using both singular and plural pronouns equally, while
Trump using more singular pronouns. The results are discussed in relation to previous lit-
erature on Red and Blue language, which refers to distinct linguistic patterns associated
with Republican (Red) and Democratic (Blue) political ideologies.

## 1 Introduction

Political debates play a crucial role in shaping the public perception of presidential candidates
and sometimes influencing electoral outcomes. Over the years, extensive research has focused
on understanding how language functions within political contexts, revealing insights into
the strategic use of words by political parties and individual candidates. This article provides a

of our data is publicly available at:
https://osf.io/sy3b7/?view_only=129698a9660a483a800674cba2b2d2ce

**Funding:** MB is funded by the European Research Council (ERC-2021-STG-101039777, project ABSTRACTION). Views and opinions expressed in the present paper are however those of the author(s) only and do not necessarily reflect those of the European Union or the European Research Council Executive Agency. Neither the European Union nor the granting authority can be held responsible for them. The funders had no role in study design, data collection and analysis, decision to publish, or preparation of the manuscript.

**Competing interests:** The authors have declared that no competing interests exist.

semantic analysis of the speech profiles of the two presidential candidates for USA 2024 elections (Donald J. Trump and Kamala D. Harris) based on the debate held on September 10th, 2024 on ABC news. We formulate research hypotheses on how candidates' lexical (semantic) and pragmatic choices can differ, and test them through quantitative and qualitative analyses. Our hypotheses and their related analyses, as well as our submitted manuscript, were formulated and conducted before election day, and therefore are unbiased by the results of the election.

Our hypotheses are based on previous findings that focused on the *language* used by the US Democrats and Republicans. We do not focus on extralinguistic factors, such as face-threatening strategies and their impact on candidate perception (as in [1–3]). We focus on linguistic factors and specifically on semantic and pragmatic aspects of the language used by the candidates, such as their word choices in terms of word concreteness and specificity. Previous research has shown that the verbal style of candidates, whether delivered through debates, campaign speeches, or social media, plays a crucial role in shaping public perceptions [4–6]. This style involves measurable semantic variables at the level of meaning of words, as well as pragmatic dynamics such as the choice of different frames (often figurative), used to express a viewpoint on given topics in such a way that they align with the strategic goals of the parties [6–8].

In relation to the US political debate, research has shown that the language used by the Democratic and Republican parties often reflects underlying ideological differences. For instance, Democratic discourse tends to emphasise compassionate and empathetic traits, while Republican rhetoric frequently highlights strength and personal responsibility [9]. This ideological divide is evident in the tone and content of debates, where each party's language strategy aligns with its broader political narrative [10,11].

The aim of this paper is to analyse the recent debate held on September 10th, 2024 between the two candidates, Donald J. Trump and Kamala D. Harris, listed in this order to reflect their positioning during the debate, from the perspective of the TV viewer (Trump on the left, Harris on the right). We seek to compare and contrast the language and the speech profiles of the two candidates overall, to explore whether their communicative styles align with what is known about language strategies used by their respective parties.

The central research question of this paper can be summarised as follows: Are the speech profiles of Trump and Harris systematically aligned with the typical characteristics identified in previous literature as peculiar to Democrats and Republicans?

We hypothesise that the speech profiles of Trump and Harris do reflect the typical characteristics associated with their parties. To test this hypothesis, we identify and list specific linguistic and pragmatic features that may differ in the language used by Trump and Harris, based on literature review; we operationalise these characteristics using different experimental methods, that combine traditional lexical and semantic resources with large language models (LLMs), supported by human-in-the-loop annotation; and finally we run a series of quantitative and qualitative analyses to compare the linguistic choices adopted during the debate held on September 10th, 2024.

Our methodology integrates both standard linguistic measures (e.g., word frequency) with state-of-the-art language model classification for political positioning (e.g., the DeBERTa algorithm for textual entailment [12], the SiEBERT [13] model, fine-tuned for polarity analysis, GPT-4o for framing analysis [14]). We employ a data-driven approach and refrain from expressing any political stance. All of our code is publicly available at: https://github.com/PhilWicke/debate_analysis24. All of our data is publicly available at: https://osf.io/sy3b7/?view_only=129698a9660a483a800674cba2b2d2ce

## 2 Theoretical background

### 2.1 Framing of values and ideology

Framing refers to the process by which certain aspects of a perceived reality are highlighted or emphasised in communication, guiding audiences toward a particular interpretation. According to [15], framing involves selecting "some aspects of a perceived reality" and making them "more salient in a communicating text," in order to promote "a particular problem definition, causal interpretation, moral evaluation, and/or treatment recommendation" (p. 52). In this way, framing shapes how information is processed and influences individuals' understanding and responses to issues. Figurative framing (e.g., [16]) is a particular type of framing that goes beyond merely selecting and presenting existing information. Figurative framing is achieved by constructing meaning by means of figurative constructs such as metaphors (a particular type of figurative language), hyperboles or metonymies.

Metaphorical framing in political discourse plays a critical role in shaping how audiences perceive complex political issues. In a thorough meta-analysis [17] illustrate how metaphors frame topics and influence public attitudes and policy preferences. For instance, metaphorically comparing the economy to a machine, one can suggest that economic policies need precise, mechanical adjustments that can be achieved because economic management is seen as a technical and controllable process. Framing immigration as a flood or wave suggests that the social phenomenon is like a forceful, potentially destructive natural catastrophe. These metaphorical frames shape perceptions by emphasizing certain aspects of an issue while obscuring others, guiding political discourse and public reaction.

In relation to the American political debate, it is suggested that Democrats often emphasise values like collective responsibility, empathy, and inclusivity [18,19]. Democrats' discourse tends to highlight the role of government in addressing inequalities and providing support to disadvantaged groups [7,20]. These values can be expressed by means of figurative frames that point to the concept of nurturing parenthood, and linguistic expressions such as "care for the vulnerable" or "protect the environment" [6].

In contrast, Republicans emphasise individual responsibility and moral authority, often stressing values like personal accountability and traditional family structure. These values can be framed using "strict father" metaphors that focus on discipline, self-reliance, law and order calls for strong, authoritative responses to crime, metaphorically represented by a strict parent enforcing rules to teach responsibility and respect [10,11,21,22].

Given these distinctions in the use of figurative framing, we hypothesise that the speech profiles of Donald Trump and Kamala Harris will reflect these ideological differences. Specifically, Trump's rhetoric is expected to emphasise personal responsibility, traditional values, and a strong stance on law and order, consistent with Republican metaphors and themes [10, 23]. Conversely, Harris' speech is anticipated to focus on collective responsibility, social justice, and the role of government in addressing societal inequalities through reforms, aligning with Democratic metaphors and themes [24].

### 2.2 Appeals to emotion

Recent studies show that the use of emotionally engaging and empathetic language can impact politicians' appeal to voters by resonating with their personal experiences and values [25]. Within the American political debate, it was recently reported that the partisan differences in moral-language use shifts over time as the parties gain or lose political power: when a party is not in power, as for instance the Democrats in the period after Donald Trump

won the 2016 presidential election, they tend to use more moral language. This is reported to be true for both parties [26–28].

Besides these time and context sensitive circumstances, research suggests that overall Democrats tend to incorporate words in their language that are loaded with emotional valence centred around positive emotions like empathy, compassion and support, in addressing social topics such as healthcare, welfare, and minority rights [20,29]. In contrast, Republican rhetoric often seems to use words that are loaded with negative valence, evoking emotions such as fear and anger, when addressing topics related to national security and immigration. The appeal to emotions such as fear and anger serves to mobilise support and underscore perceived threats to national interests [10,23,30]. We therefore expect Trump's language to be characterised by words that show a stronger negative valence compared to Harris, and vice versa, Harris' language is expected to be characterised by words that show a stronger positive valence compared to Trump.

Another aspect associated with the evaluation of the overall sentiment encoded in words, is their so-called "subjectivity" [31]. In text analysis, the subjectivity variable quantifies the amount of personal opinion and factual information contained in the text. Higher subjectivity means that the text contains many words expressing personal opinion rather than words expressing factual information or supposed to express factual information. In political discourse, this distinction can be interpreted as follows: low subjectivity scores may suggest that statements are expressed in such a way to appear as objective facts, while high subjectivity scores may indicate that statements explicitly reflect the candidate's personal opinion and political stance on a given issue. Based on this idea, we explore the candidates' average subjectivity scores.

Regarding the emotional dimension of discourse, we briefly introduce foundational work on psychosocial constructs underlying word choices (and for a broader perspective on the use of natural language processing to understand people and culture, we refer to [32]). Notable here is the work by Pennebaker et al. [33] with extensive research contributions - which demonstrates that subtle aspects of language use (such as function word and pronoun usage) serve as reliable indicators of underlying psychological states and social dynamics [34, 35]. This literature not only informs our analysis of political discourse but also provides a broader framework for understanding the cognitive and affective processes reflected in language, which is also reflected in other works [36,37]. Boroditsky's work, for instance, compellingly shows that the structure of language can shape thought and perception, thus offering an important complement to our discussion. In line with this is recent experimental evidence [38], which directly tests how subtle linguistic metaphors—such as those framing immigration—can systematically influence attitudes.

## 2.3 Policy presentation: details vs. big picture

In his classic essay on *Politics and the English Language* [39], almost a century ago the British writer George Orwell argued that two of the most significant problems in political language are "staleness of imagery" and "lack of precision," which lead to a loss of clear communication (staleness of imagery) and vagueness that is used to conceal inconvenient truths (lack of precision). Orwell emphasises that the remedy to such deceptive language lies in choosing words that are clear (concrete) and precise (specific). Note that concreteness and specificity are two different variables involved in the notion of conceptual abstraction [40], where concreteness defines how tangible a concept is ("wall" is highly concrete, while "protection" is more abstract) and specificity defines the lexical level at which a concept is expressed ("stone wall" is highly specific while "barrier" is more general).

Recent analyses of political debates indicate that lexical precision plays a significant role in shaping the public's perception of candidates' competence and trustworthiness [41]. In relation to the American political discourse, research suggests that Democratic candidates indeed tend to present policies with a high degree of word specificity, providing detailed explanations [42]. Right-wing parties instead tend to emphasise broad themes and big-picture narratives in their policy presentations. Their language can be described as more punchy and straightforward, designed to clearly communicate key points by means of clear-cut, impactful statements [43]. Their preference for less technical language may be interpreted as an attempt to frame issues in relatable, everyday terms, avoiding the complexities that can alienate voters who are less familiar with policy complexities.

Given these rhetorical strategies, we hypothesise that Harris will use words associated with higher levels of specificity, while Trump will employ words associated with lower levels of specificity.

## 2.4 Linguistic style: complex vs. direct

Building on the variables outlined above and reflecting on Orwell's classic critique of political language, the so-called "staleness of imagery" essentially refers to word concreteness. Word concreteness, namely the use of words that designate tangible referents, plays a vital role in shaping how messages are received and understood [44].

In political discourse, Democratic candidates appear to often employ rather abstract words, reflecting their focus on addressing systemic issues and nuanced perspectives [45]. This rhetorical style, while intended to appeal to an educated and progressive audience, can result in messages that are less direct and potentially harder for some listeners to follow.

In contrast, Republican candidates often favour simpler, more direct language that is characterised by the use of concrete words that designate tangible referents, such as in the slogan "Build the Wall" or "Lock her up", to ensure messages are easily understood and resonate with a broad audience. Research shows that this kind of concrete language is particularly effective in political debates, as it enhances memorability and facilitates voter engagement by providing vivid imagery that people can easily recall and -possibly- relate to [45,46].

Given these rhetorical strategies, we hypothesise that Harris' language may use more abstract words compared to Trump who, in turn, may leverage concrete words to reach a wider audience.

## 2.5 Identity politics and group appeal

Democratic candidates frequently utilise rhetoric that targets specific identity groups, such as minorities, women, LGBTQIA+ communities, and the working class. Their discourse emphasises inclusivity and diversity, portraying the government as an advocate for equality and social justice. This can be reflected in the use of inclusive pronouns like "we" to foster a sense of collective belonging and shared goals [47].

A different approach involves the use of more first-person singular pronouns, such as "I" and "me", to emphasise individual responsibility, authority, and personal agency. First-person singular pronouns evoke self-focus and self-presentation, allowing speakers to express their viewpoints and assert control over their narrative. This rhetorical choice aligns with the conservative values of self-reliance and personal accountability. By using "I" pronouns, speakers position themselves as strong, decisive leaders who take direct action to address problems, which resonates with their base's preference for assertive, authoritative figures [48].

Given these rhetorical differences, our hypothesis is that Harris' language will show a higher frequency of inclusive pronouns like "we" and "us" to emphasise collective action and

shared identity, while Trump's language will include more individualistic pronouns such as "I" and "me" to highlight personal authority and decisiveness. In addition to our primary analyses, we will also explore the usage of direct references to the opposing candidate in the political debate. This will allow us to identify potential patterns or rhetorical strategies that may emerge, providing insight into the role of direct references in shaping the dynamics of political exchanges.

The complete list of variables and methods used to address their study are hereby summarized in the scheme below.

| Study | Measure | Result |
|---|---|---|
| Framing of Values and Ideology | Framing Identification & Analysis (3.1) | Section 3.2 |
| Appeals to Emotion and Logic | Subjectivity / Polarity (4.1) | Section 4.2 |
| Details vs. Big Picture | WordNet's lexical hierarchy (5.1) | Section 5.2 |
| Complex vs. Direct | Brysbaert concreteness scores (6.1) | Section 6.2 |
| Identity Politics & Group Appeal | Political DEBATE (7.1) | Section 7.2 |

## 3 Study 1: framing of values and ideology

### 3.1 Methods

We based our analysis on the transcripts of the debate aired on ABC News, obtained from abcnews.go.com (https://abcnews.go.com/Politics/harris-trump-presidential-debate-transcript/story?id=113560542).

First, the study is an analysis of the most frequent words, providing a preliminary overview of the key terms used by each candidate in both debates. This serves as an indicator of potential trends, which are then further explored in the second part through a more detailed framing analysis in relation to the two major US political parties.

*3.1.0.1 An overview of the debate*

The candidates' replies to the moderator can be summarised as follows: Trump provided a total of 74 responses, while Harris contributed 34. Despite giving fewer responses, Harris' replies were, on average, longer than Trump's, with an average of M=173.79 words per response compared to Trump's M=109.36. Similarly, Harris' responses exhibited a slightly higher average word length (M=4.55 characters) than Trump's (M=4.32 characters). When considering the total number of words, Trump used 8,093 words (tokens) across all his responses, whereas Harris used 5,909. The diversity in their word choices is reflected in the total number of unique words (or word types) used: Trump employed 1,745 unique words, while Harris used 1,611. Notably, Harris demonstrated a higher average number of unique words per response (M=47.38) compared to Trump's M=23.58. Lastly, both candidates exhibited similar sentence length, with Trump averaging M=10.69 sentences per response and Harris close behind at M=10.35 sentences per response. These descriptive statistics provide an initial view of the linguistic tendencies of both candidates during the debate. To provide an accessible overview of the frequency data, we include word clouds and graphs that visually represent the most common terms. For our frequency analysis, we perform two preprocessing steps: (i) lemmatisation—the process of reducing words to their base or dictionary form, known as lemmas—using the SpaCy library to retrieve the lemmas (https://spacy.io/api/lemmatiser); and (ii) the removal of stop words from the text using the NLTK stop word list (NLTK Stop Word List). The word clouds in Fig 1 have been generated using the Python wordcloud package (https://github.com/amueller/word_cloud).

The bar graphs in Fig 2 show the top 20 most common words across all candidates' responses, after lemmatisation and stop words removal.

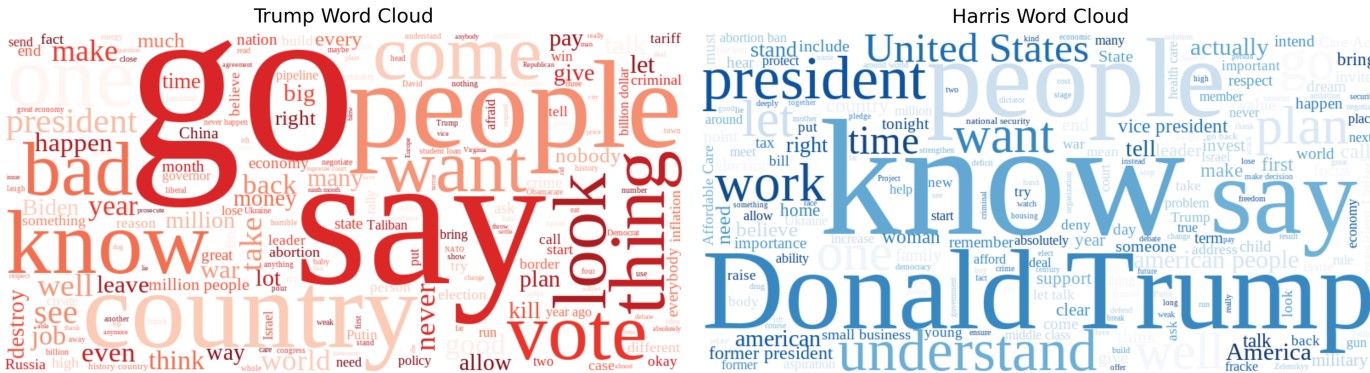

**Fig 1. The word cloud shows red words for Trump and blue words for Harris, with word size representing frequency, highlighting the most common terms in their responses.**

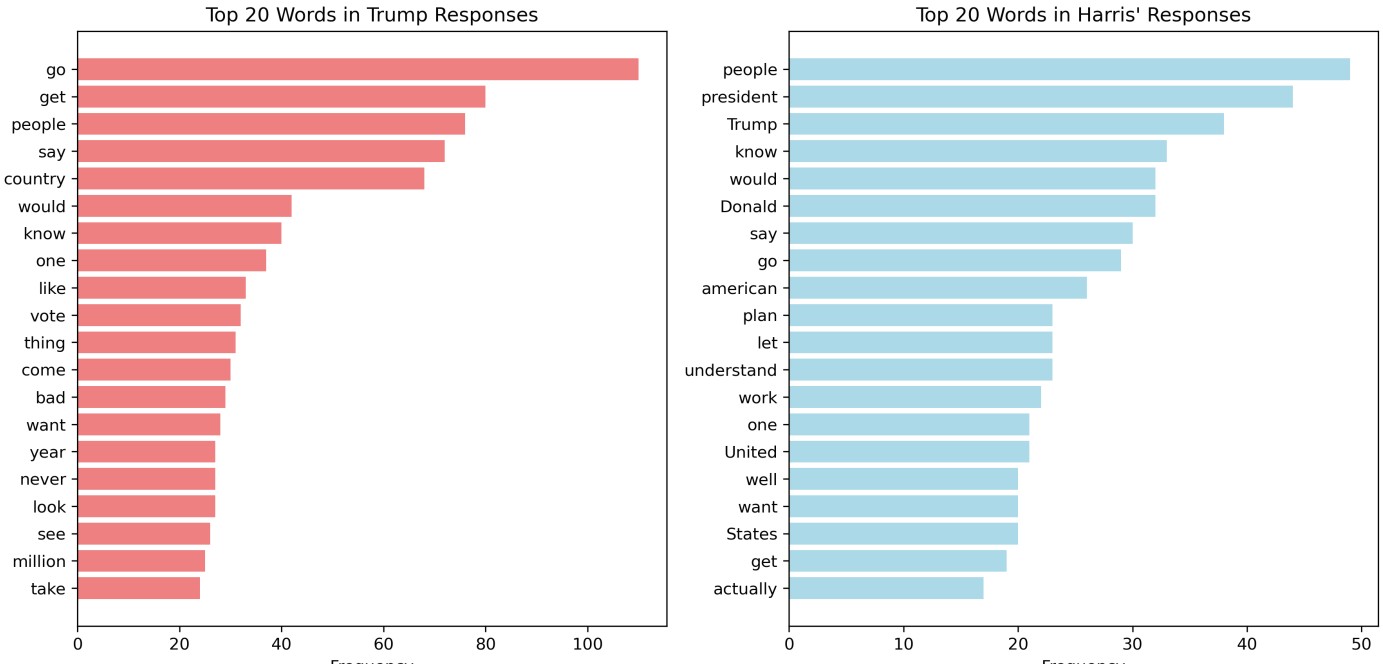

**Fig 2. Bar plots showing the frequency of words in both candidates' responses.** Left/Red plot shows Trump's most frequent words with "go" being by far the most frequent word. Right/Blue shows Harris' most frequent words with "people", "president" and "Trump" as the top three most frequent words.

Specifically, we also identified the top five most frequent words from Trump's responses that were not present among the 100 most frequent words in Harris' responses, and vice versa, arguing that this approach isolates the most distinctive terms used by each candidate. For Trump, the top five unique words are: "like" (33 occurrences), "vote" (32), "thing" (31), "bad" (29), and "never" (27). In contrast, Harris' top five distinctive words are: "Donald" (32 occurrences), "American" (26), "understand" (23), "work" (22), and "States" (20). Since many of these most frequent words are action words, we also isolate named entities in order to investigate people and institutions that each candidate might refer to. We use the SpaCy named entity recognition to provide an overview visualised in Fig 3.

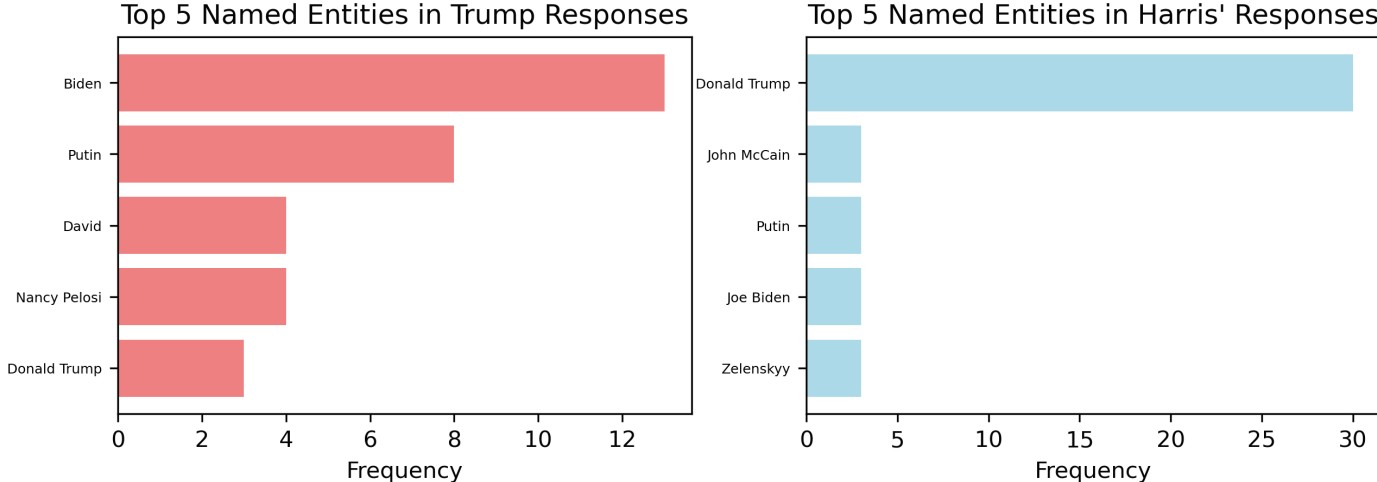

**Fig 3. Top ten frequencies of named entities in Trump (red) and Harris' (blue) responses.** With "Donald Trump" being Harris' most mentioned entity and Biden being Trump's most named entity.

*3.1.0.2 Framing analysis*

Framing analysis is inherently complex and time-consuming, especially when dealing with figurative language, as it requires a nuanced understanding of context and meaning and extensive manual annotations. To address this challenge, we incorporate state-of-the-art computational tools to assist in the identification and classification of frames. Specifically, we rely on the GPT-4o model [49] developed by *OpenAI*, one of the most advanced large language models (LLMs) currently available. The study in [14] demonstrates the use of LLMs for automated framing analysis. While LLMs have shown promise in tasks requiring interpretative judgments, we approach the use of GPT-4o cautiously, mindful of the inherent risks in relying solely on automated systems for such a nuanced task [50].

To mitigate these risks and ensure the robustness of our analysis, two linguistically trained experts—the authors of this paper—conducted an independent manual annotation of the frames identified by GPT-4o, aimed at identifying false positives (i.e., frames identified by the LLM that are not actual figurative frames) and false negatives (i.e., frames missed by the LLM). This dual approach, combining human expertise with automated model outputs, allowed us to critically evaluate the frames and ensure the reliability of the results. The identified frames were then qualitatively analysed, grouped into broader categories, and examined for trends and clusters within the data.

**Framing Annotation Process:** The first step in our process involved querying the GPT-4o model [49] through the *OpenAI* API. The prompt used for this task is:

```
Given the following response in a debate, does the response
make use of figurative frames?
   If so, say yes and list the figurative frames with their
explanation in brackets. Do not use new lines for your
response. If no figurative frames were used, say 'no figura-
tive frames'. Response:
```

The model was set to a temperature of 0, ensuring deterministic outputs for consistency. We limited the response length to 250 tokens, balancing comprehensiveness with relevance. Development of the prompt is based on experience through unrelated LLM experiments [51].

This prompting procedure was applied to both the responses from Trump and Harris during the debate.

Once the model generated its responses, we compiled the results into datasheets for further analysis. The two annotators independently assessed each of the model's identified frames, labelling them according to the following categories:

- **Correctly Identified**: The annotator agrees with the model's identification of a figurative framing in the response.
- **Falsely Identified**: The annotator does not consider the identified pattern to constitute a figurative frame.
- **Not Identified**: The annotator identifies a figurative framing that the model has missed.

Following the manual annotation, we applied an algorithm to assess patterns and evaluate consistency in the model's framing judgments: The annotation data from two annotators was relabelled into a numerical format for structured comparison. The three labels ("Correctly Identified" etc.) were mapped to specific numeric values. If an item was "Correctly identified", it was assigned the value 1, while "Falsely identified" was assigned the value 2. In cases where an item was "Not identified", an additional check was made, whether the frame that was not identified was agreed upon by the annotators. If there was agreement on the missing frame between the annotators, the item received the value 3 for both; if not, it was assigned either 4 or 5, depending on the annotator. In case of discrepancies in the number of annotations between the two annotators, missing annotations were filled in with the value 6 to ensure both datasets remained comparable. This process resulted in two numeric lists representing the annotation outcomes in a consistent format, facilitating a direct comparison of the annotators' decisions.

**Evaluation of annotator agreement:** To quantify the level of agreement between the annotators, we calculated Krippendorff's alpha, a robust statistical measure used for assessing the reliability of annotators in coding qualitative data [52]. We used the Python package *krippendorff* [53] for the evaluation. Krippendorff's alpha is particularly well-suited for this analysis because it can accommodate any number of annotators, different types of measurement scales, and missing data.

## 3.2 Results

A total of 314 framings were annotated. Given the large number of framings, we publish the results along with the annotation data in our online repository (https://osf.io/sy3b7/?view_only=129698a9660a483a800674cba2b2d2ce).

In the evaluation of annotator agreement, Krippendorff's alpha scores indicated a high level of reliability between the two expert annotators. For Harris' responses, the alpha score was $\alpha = 0.887$, demonstrating a near-perfect agreement. This strong consistency underscores the robustness of our annotation process and suggests that the framing patterns in Harris' statements were readily identifiable. For Trump's debate responses, the alpha score was $\alpha = 0.830$, reflecting also in this case a very high agreement in the identification of figurative framing. This score suggests that while there were occasional differences in interpretation, the annotators largely concurred in their judgements of the frames present in Trump's rhetoric. The analysis shows that in 62.10% of cases both annotators agreed that the framing identified by the LLM was correct. In an additional 28.66% of cases, one annotator agreed while the other did not, meaning that in 90.76% of instances the LLM's output was validated by at least one annotator - and 9.24% of framings noted falsely identified. We believe these findings

highlight the robustness of the LLM in capturing relevant framing, even accounting for some of the inherent subjectivity in human judgment.

After resolving the disagreements between the two annotators through discussion, we proceeded to analyse the final annotations.

The total number of identified uses of figurative frames across all responses for Trump is 111 and for Harris is 93. Given the 74 responses by Trump, he uses figurative frames 1.5 times per response, whereas Harris uses about 2.74 frames per response. Notably, there is a series of responses for both candidates that provide no informational, but rhetorical value such as:

> **Harris**: Come on.
> **Trump**: Would you do that? Why don't you ask her that question –
> **Harris**: Why don't you answer the question would you veto –
> **Trump**: That's the problem. Because under Roe v. Wade.
> **Harris**: Answer the question, would you veto–

**Trump's figurative frames** reveal distinct themes at both content and framing levels, which can be grouped and labelled into three overarching categories:

- **Nation in decline and external threats:** On the content level, Trump appears to frame the country as facing existential decline, using metaphors like "we're a failing nation," "our country is being lost," and "they're destroying our country." Immigration in particular is framed as one of the main problems, and it is framed as an invasion or takeover by immigrants, who are responsible for "taking over the towns" and "pouring into our country." The framing relies heavily on hyperboles and metaphors to convey catastrophic outcomes, which are used to evoke strong emotional responses. For example, terms like "bloodbath" and "World War 3" amplify the gravity of the threats, while hyperboles like "the most embarrassing moment in the history of our country" intensify the stakes.
- **Economic disintegration and betrayal:** Economic matters are framed as both internal and external betrayals. Internally, economic decline is represented through hyperbolic statements like "they will kill the United Auto Workers" and "selling our country down the tubes." Externally, the narrative of unfair trade and exploitation is emphasised with metaphors like "ripping us off" and "being ripped off by NATO." The framing strategy here mixes hyperbole and metaphor, invoking strong, clear images of economic betrayal and loss. This leads to a portrayal of the nation's economy as victimised by external forces and incompetent domestic policy.
- **Moral and social decay:** Trump also appeals to a frame of moral decline and social disorder. Crime is repeatedly framed with metaphors like "crime through the roof" or "new form of crime," while broader societal concerns are escalated using metaphors such as "torn our country apart." On a moral level, phrases like "execute the baby" and "weaponized justice" are designed to provoke emotional reactions and present the opposition as threatening moral and social values.

Across all categories, the content level paints a picture of a nation in existential danger, facing economic destruction, external threats, and moral collapse. This is shaped by intense hyperbole, metaphor, and emotionally charged language, intended to stir strong emotional responses of fear, betrayal, and urgency.

**Harris' figurative frames** suggest themes centred on recovery, empowerment, and unity, contrasting with Trump's focus on decline and existential threats. The frames can be grouped into three main categories:

1. **Economic empowerment and opportunity:** On the content level, Harris frames economic recovery and growth through metaphors like "lifting up the middle class," "opportunity economy," and "backbone of America's economy." These expressions emphasise a positive, forward-looking approach to strengthening the economy, focusing on middle-class support and long-term stability. The economy is personified as a system that can be improved, grown, or supported. This framing conveys optimism and inclusivity, using empowering language such as "giving hard-working folks a break" and "building a clean energy economy."

2. **Restoring and defending the nation's values:** The content here is framed around defending the country's core values, reflected in phrases like "clean up Donald Trump's mess," and "stand for democracy." Metaphors such as "clean up" and "turn the page" suggest moving beyond past mistakes and planning for a better future.

3. **Unity and national strength:** Content-wise, this theme centres on the unification of the nation and strengthening of its international standing. Phrases such as "bringing us together" and "chart a course for the future" frame Harris as a leader focused on collective unity and strategic direction. This framing invokes metaphors of collaboration and planning, as seen in "chart a new way forward" and "uphold the strength and the respect." Through these words, Harris aims to project stability, emphasising a vision of a united and forward-looking America.

Overall, Harris' framing strategy focuses on economic opportunity, recovery from past mistakes, and unity. On the framing level, the focus is on empowerment, restoration, and national pride.

## 4 Study 2: Appeals to emotion and logic

### 4.1 Methods

Our sentiment analysis looks at the polarity (negative/positive valence) and subjectivity (subjective/objective) of the responses given by the candidates during the debate. First, we employed the SiEBERT [13] model, which is specifically fine-tuned on polarity analysis. This model applies binary polarity classification to various types of English-language texts. We tracked the distribution of polarity for each response across the debate for Trump and Harris. Each response contained at least one or more sentences for which we retrieved the negative or positive classification. This level of granularity allowed for a more detailed understanding of polarity variations across individual sentences.

We averaged over all sentences for each response to estimate a **polarity** value. This polarity value was then linearly transformed to the range of -1 (negative) to 1 (positive). After evaluating all polarity scores, a Shapiro-Wilk test was performed to assess the normal distribution of both polarity vectors. Depending on the results of the normality test, either a Kolmogorov-Smirnov or Mann-Whitney U test was conducted.

Recognizing the possibility of sentence dependence within responses, we acknowledge that some interdependence may exist between sentences, but argue that this dependence does not diminish the value of the statistical test, as political rhetoric often varies greatly sentence by sentence, making sentence-level analysis appropriate for capturing these shifts. Political speeches and written rhetoric are often deliberately designed with varied sentence-level constructions. Politicians may use one sentence to assert an opinion followed by another that presents a fact, or even interrupt a narrative to introduce a contrasting perspective. In such texts, the meaning and tone can change rapidly from sentence to sentence. This variability is consistent with findings in discourse analysis where authors use shifting rhetorical strategies

to engage different audiences or to emphasize contrasting points [54,55]. An example of this can be seen in the following statement by Trump: "But they still bring it up just like they bring 2025 up. They bring all of this stuff up. I ask you this. You talk about the Capitol. Why are we allowing these millions of people to come through on the southern border?" In this statement, Trump shifts from referencing the January 6 insurrection ("they still bring it up") to Project 2025 ("like they bring 2025 up"), then to a more general criticism of his critics ("They bring all of this stuff up"), and finally to a concern over border security ("people to come through on the southern border").

In a second analysis, we used two models to assess **subjectivity**. First we used the *spaCyTextBlob* library (https://spacy.io/universe/project/spacy-textblob) with the `en_core_web_lg` model to analyse the subjectivity of the responses. Analogous to the polarity score, we computed the subjectivity score as an average across all sentences within each response and across all responses. Subjectivity measures how much a statement is opinion-based, with values from 0 (objective) to 1 (subjective). We performed statistical tests to compare the degree of subjectivity between the two candidates in the same way described above for the polarity analysis. We then used a second model to replicate and validate the subjectivity results of the *spaCyTextBlob* model. In particular, we used a fine-tuned mDeBERTa V3 model for subjectivity detection [56] (https://huggingface.co/GroNLP/mdebertav3-subjectivity-english). This model, called *Thesis Titan*, was developed as part of the CLEF 2023 Task 2 [57], and is specifically designed for classifying newspaper sentences as either objective or subjective. The underlying assumption of this model is consistent with that of *spaCyTextBlob*: a sentence is classified as subjective if it is influenced by personal feelings, opinions, or tastes, whereas objective sentences are grounded in factual information and lack personal bias [58].

## 4.2 Results

**Polarity Analysis** based on the SiEBERT [13] model provides the polarity scores depicted in Fig 4. If a response has a value at -1, this indicates that one or all of the sentences in the response were classified as having negative sentiment by the model. Overall, the polarity of the language used by both candidates is quite negative. In particular, Trump's responses have an average negative sentiment of: M = -0.351 (SD = 0.568), whereas Harris' average is M = -0.167 (SD = 0.602). The Shapiro-Wilk test for Harris' scores is not providing reliability for normal distribution (p=0.069) and the Kolmogorov-Smirnov test also indicates no significance in the difference between the two profiles (p=0.314). Therefore, the distribution of polarity scores of the language used by the two candidates do not significantly differ.

**Subjectivity analysis** results of the *spaCyTextBlob* model are depicted in Fig. 5, which visualise the average subjectivity of Trump's and Harris' responses, respectively. Bars are color-coded from light (objective response) to dark (subjective response) to indicate increasing levels of subjectivity, providing a clear representation of each response's objectivity.

In particular, Trump's responses have an average subjectivity of: M = 0.202 (SD = 0.310), whereas Harris' average is M = 0.218 (SD = 0.274). The Shapiro-Wilk test for both scores is not providing reliability for normal distribution (p > 0.05), see Table 1 for statistical results. The Kolmogorov-Smirnov test indicated a significant difference in subjectivity between the two candidates (p <.001). Therefore, the distribution of polarity scores of the language used by the two candidates do significantly differ.

These results are in line with those of the fine-tuned mDeBERTa V3 model for subjectivity detection. The subjectivity scores of this model show a mean of M = 0.414 (SD = 0.287) for Trump and a mean of M = 0.583 (SD = 0.249) for Harris. The Shapiro-Wilk test indicated

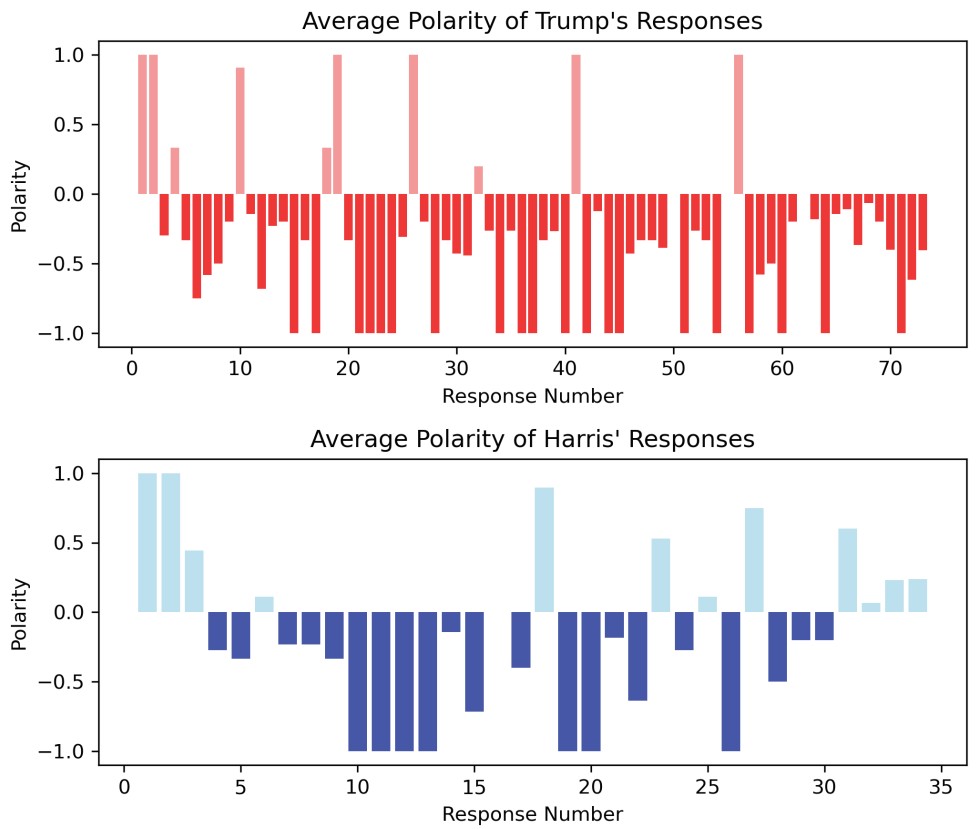

**Fig 4. For each response we average the amount of positive (1) and negative (-1) sentences to retrieve the distribution of sentiments across the debate for Trump (red) and Harris (blue).** A large bold bar at -1 therefore indicates that one or all of the sentences in the response were classified as negative by the model.

that Trump's subjectivity (W = 0.907, p<.001) and Harris' subjectivity (W = 0.922, p = 0.020) deviated from normality (see Table 1). The Kolmogorov-Smirnov test showed a significant difference (p = 0.005) between the distributions. Therefore, even if the models show deviating values, which is due to their different architecture and training, both provide evidence that Trump is slightly, but significantly, less subjective than Harris.

#### 4.2.0.1 Qualitative results

In addition to the quantitative analysis, we examined some of the most negative sentences from the responses of Trump and Harris. We use the sentiment classifier to collect the responses that have the greatest number of negative sentences and the greatest number of positive sentences. For Trump, the most negative responses, exemplified and interpreted in the next paragraph, are: #19, #51, #2, #48 and #73. And the most positive responses are: #2, #67, #51, #3 and #9. We see that #2 and #51 contain a large number of negative and positive sentences. For Harris, the most negative responses are: #29, #21, #13, #31 and #20. And the most positive responses are: #22, #2, #33, #31 and #17. We see that #31 is the only response to contain many negative and positive responses.

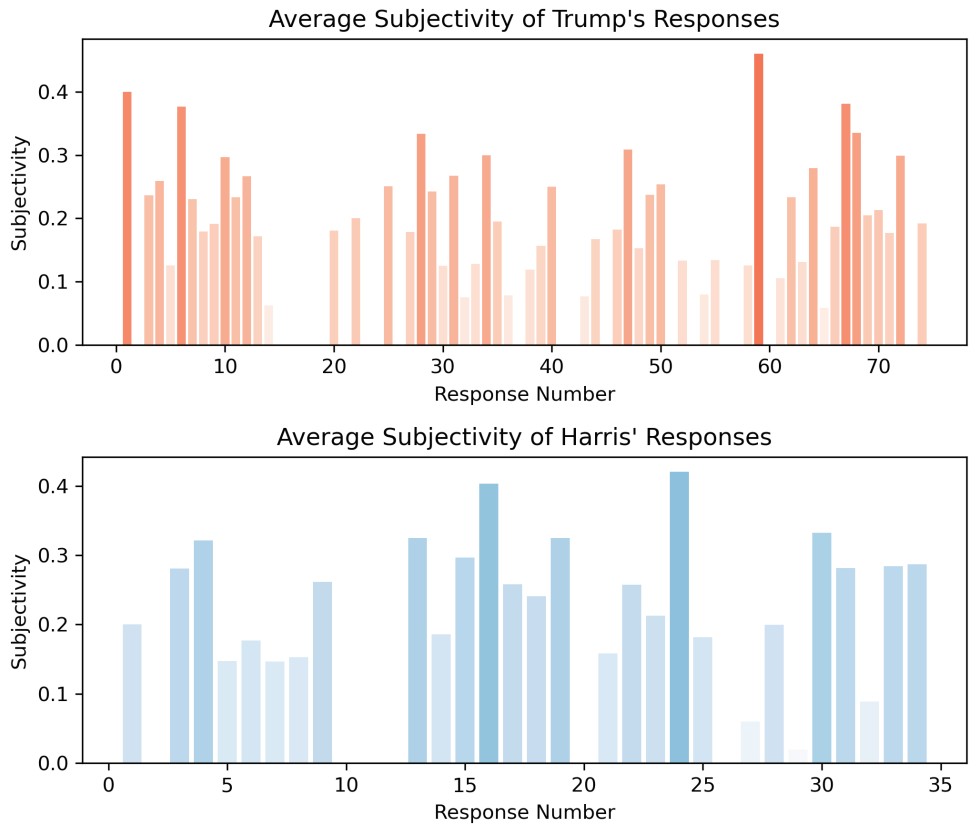

**Fig 5. The upper bars (red shades) indicate the average (across sentences) subjectivity score of Trump's responses.** The lower bars (blue shades) show those for Harris' responses. A bold bar with high subjectivity indicates that the response contained many subjective sentences. These subjectivity scores are evaluated by the *spaCyTextBlob* model.

**Table 1. Statistical results of polarity and subjectivity evaluation.** ∗ **indicates statistical significance**

| Test | Model | Metric | Statistic | P-value |
|---|---|---|---|---|
| **Shapiro-Wilk Test** | *spaCyTextBlob* | Trump's Subjectivity | 0.7008 | $1.46 \times 10^{-36}$∗ |
| | *spaCyTextBlob* | Harris' Subjectivity | 0.7930 | $8.04 \times 10^{-22}$∗ |
| | *Thesis Titan* [56] | Trump's Subjectivity | 0.9069 | $4.19 \times 10^{-5}$∗ |
| | *Thesis Titan* [56] | Harris' Subjectivity | 0.9217 | $2.04 \times 10^{-2}$∗ |
| **Kolmogorov-Smirnov Test** | *spaCyTextBlob* | Subjectivity | 0.1214 | 0.0007∗ |
| | *Thesis Titan* [56] | Subjectivity | 0.3539 | 0.0045∗ |

**Trump's Polarity** is analysed comparing the most positive and most negative responses. A clear distinction emerges in their linguistic patterns: The most positive responses generally feature more confident and assertive language, with a tone of accomplishment and control. For instance, these responses often emphasise achievements, using phrases like "I created one of the greatest economies," "we did a phenomenal job," (#2) and "I will get it settled before I even become president" (#51). This language conveys a sense of certainty and self-assurance, underscoring the Trump's role as an active agent in solving problems, managing crises, or bringing about significant changes. Furthermore, there is a strong focus on collective action and leadership, as seen in statements like "we were able to do that" (#9) and "they respect your president" (#51). Discussing challenges, the tone remains optimistic, with

promises of improvement: "I'll do it again and even better" (#2) or "if we can come up with a plan… I would absolutely do it" (#67).

On the other hand, the most negative responses display a more critical and adversarial tone, often emphasising failure and decline. Words such as "failing," "destroying", and "disaster" appear frequently, painting a bleak picture of the current state of affairs. The negative responses focus heavily on blaming others, particularly political opponents, as seen in phrases like "she's destroying this country" (#19) or "they don't have the courage to ask Europe" (#51). The language is infused with urgency and fear, with references to catastrophic outcomes such as "World War 3" or "we're going to end up in a third World War" (#73). Trump often contrasts his own strength or achievements against the perceived weakness of others: "I know Putin very well. He would have never…" (#51) or "if I were president it would have never started" (#48).

Overall, the linguistic pattern of the most positive responses revolves around confidence, leadership, and forward-looking optimism, while the most negative responses are marked by blame, fear, and a sense of impending disaster. Both sets of responses are highly rhetorical, but the former seeks to inspire confidence, whereas the latter focuses on stoking concern and dissatisfaction.

**Harris' Polarity** reflects a tone of empathy and inclusion in the positive responses. For instance, in response 22, Harris balances support for Israel's right to defend itself with concern for Palestinian civilians, using phrases such as "innocent Palestinians have been killed" and "we need a cease-fire deal". This reflects a diplomatic tone, recognising the humanity on both sides of a conflict. Her choice of positive words reflect optimism: In #33, she focuses on hope, unity, and a vision for a better future: "I believe in what we can do together" and "we can chart a new way forward". This reflects a belief in collective action and a bright future, with a contrast to past approaches.

Her negative responses concentrate heavily on the shortcomings and divisive actions of Trump. For example, #29 criticises the Trump's historical actions with phrases like "attempted to use race to divide the American people" and references to racially charged incidents such as the Central Park Five and birtherism. The negative responses also take a moral stance, portraying Trump as unfit for leadership. In #21, Harris references military leaders calling Trump "a disgrace," and in #31, Harris accuses Trump of "continuous lying".

**Trump's subjectivity** is analysed by looking at the five most subjective and five most objective responses in a qualitative manner. In Trump's most objective responses, such as response #37, the tone is centred around factual claims related to Trump's role or lack thereof in a particular event, alongside explicit mentions of evidence or documentation. For example, Trump references a tape from Nancy Pelosi's daughter and provides an account of offering National Guard support, which was allegedly rejected: "I said I'd like to give you 10,000 National Guard or soldiers. They rejected me. Nancy Pelosi rejected me. It was just two weeks ago, her daughter has a tape of her saying she is fully responsible for what happened." This factual recounting of past actions shows characteristics of objectivity that a prediction model rates as objective statements.

Similarly, in response #28, Trump discusses ongoing legal cases and the concept of "weaponization" in a factual tone. While Trump conveys a defensive stance, the mention of Supreme Court decisions and explicit descriptions of the political cases adds to an objective perception. The repetition of the phrase "it's weaponization" highlights a consistent attempt to frame the issue within legal and institutional terms. The frequent use of "they" versus "I" in these responses (e.g., "they weaponized the justice department") also reinforces this distance from personal bias, at least in tone.

However, in the subjective responses, particularly response #2, Trump shifts toward a more exaggerated, hyperbolic narrative. The speaker introduces scenarios like "millions of people pouring into our country from prisons and jails" and catastrophic economic decline with phrases such as "the worst in our nation's history". These extreme phrases are subjective, as they are not only emotional but are difficult to quantify or substantiate through objective evidence. The imagery of "people taking over towns" and "eating the pets" (#19) further escalates the emotional intensity. This heightened emotional appeal suggests a subjective, rather than fact-based approach within the figurative framing (Section 3.1).

In response #51, Trump invokes personal relationships with world leaders, notably Zelenskyy and Putin, as evidence of their capacity to resolve international conflicts. The claim "I will get it settled before I even become president" reflects subjective overconfidence, again leaning on self-referential authority and broad assumptions rather than grounded claims.

Overall, while Trump does present objective statements tied to concrete events and actions, their subjective responses demonstrate a preference for emotionally charged language. This distinction between calmer, factual recounting of events and extreme portrayals of danger or failure may explain why a prediction model might classify the Trump as more objective overall, but still prone to subjectivity in moments of heightened emotion.

**Harris' subjectivity** is analysed by looking at the top 5 most subjective and objective responses. In subjective responses, such as response #22, Harris introduces emotionally charged language, describing violent and tragic events like the October 7 Hamas attacks on Israel. Words like "slaughtered," "horribly raped," and references to the suffering of civilians (both Israelis and Palestinians) create an emotional narrative. Additionally, the Harris makes normative statements, like advocating for a cease-fire and a two-state solution, which reflect personal values and opinions. This combination of descriptive imagery and moral argument contributes to a subjective tone.

Responses #20 and #31 show similar subjective characteristics, particularly in the framing of January 6th and issues like healthcare. In both cases, the speaker employs emotionally resonant language, emphasising the need to "end the chaos" or standing against "continuous lying." The use of personal recollections, like mentioning John McCain's stand on healthcare or the violence on January 6th, adds personal weight and emotional charge, amplifying the subjectivity of the message.

In contrast, objective responses from Harris often lean on factual recounting of events or policy proposals. For example, in response #2, Harris outlines specific economic plans, such as extending tax cuts for families and providing support for small businesses, offering a detailed plan without emotional rhetoric. The factual structure of this argument makes it more objective.

## 5 Study 3: policy presentation: details vs. big picture

### 5.1 Method

We used the same pre-processed data as in Sec. 3.1 (lemmatised and without stopwords) to assess the specificity of the responses. To quantify the lexical specificity of words in the responses, we used *WordNet* [59], a lexical taxonomy of the English language, to determine each word's depth in the lexical hierarchy. In WordNet, depth in the lexical hierarchy refers to the number of hierarchical steps (or levels) between a given word and the root of its semantic category. This depth represents how specific or general a word is within the hierarchy. For example, in WordNet's noun hierarchy, "animal" is at a shallower depth than "dog," which in turn is shallower than "Golden Retriever." The deeper a word is in the hierarchy, the more specific its meaning.

To address potential issues with polysemy, we selected the most frequent synset (the first returned by *WordNet*) for each word, which helps mitigate ambiguity in meaning. Our analysis focused exclusively on content words—nouns, verbs, adjectives, and adverbs—by employing part-of-speech tagging to filter out function words such as prepositions and conjunctions. This decision ensures that the analysis emphasises semantically rich vocabulary, thereby enhancing the robustness of our findings.

We excluded from the analysis words that are not present in *WordNet* to prevent artificially low specificity scores, maintaining the integrity of our average specificity measurements. Each response's average specificity score was computed by summing the depths of the identified content words and normalizing this sum by the number of such words. This method provides a balanced comparison of lexical specificity between responses while controlling for sentence length and reducing the influence of high-frequency, low-information words. By employing this nuanced approach, we aimed to capture the richness and specificity of language used by the candidates.

In line with the previous analysis in Sec. 4, we evaluated the results on two levels: on the word level and on an average level of specificity of the entire response. The retrieved specificity distributions are compared for both speakers in order to assess whether or not there is a statistical significance between the specificity of words. Again, we first conduct a Shapiro-Wilk test to check for normality, followed by a Kologorov-Smirnov test given that normality cannot be assured.

## 5.2 Results

Fig 6 depicts the distribution of average specificity scores across for each of Trump's 74 responses (red/left) and Harris' 34 average specificity score per response (blue/right).

To examine the differences in lexical specificity between the responses of Donald Trump and Kamala Harris, we conducted statistical tests on the specificity scores assigned to each word in their respective responses. Initially, we assessed the normality of the specificity score distributions using the Shapiro-Wilk test. The results indicated that the specificity scores for Trump (SW = 0.815, p = 0.022) and Harris (SW = 0.832, p = 0.002) were not normally distributed.

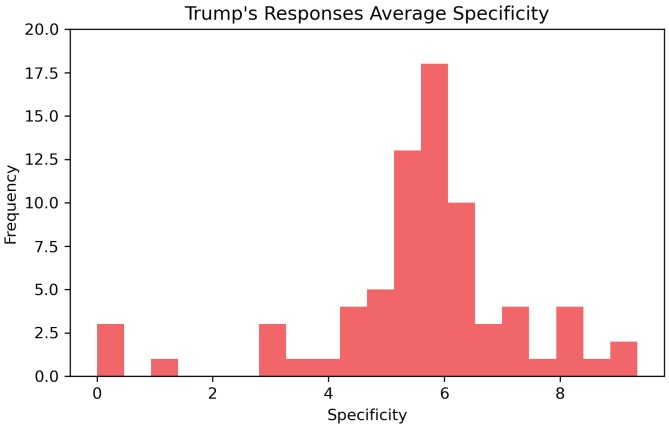
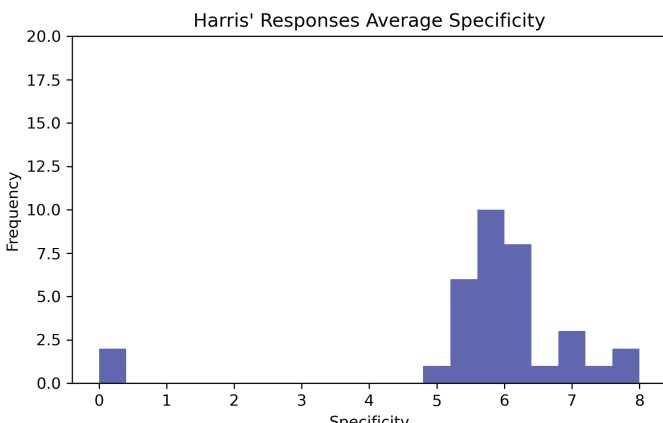

**Fig 6. Histograms depict the distribution of average specificity scores across the responses of Trump (red/left) and Harris (blue/right).** Notably, Trump responded 74 times and Harris 34 times, which leads to much lower frequencies for specificity.

Given the non-normality of the distributions, we employed the Kolmogorov-Smirnov test to compare the specificity score distributions of the two candidates. The test yielded a KS = 0.414 (p = 0.150) statistic. The high p-value indicates that there is no significant difference in the specificity score distributions between Trump and Harris. This finding suggests that both candidates used language with comparable levels of lexical specificity in their responses, despite the differing rhetorical styles typically associated with their speeches.

## 6 Study 4: linguistic style analysis – complex vs. direct

### 6.1 Method

To operationalise the complexity of the language used by the two candidates we employed two complementary approaches: a study based on word concreteness, in which we utilised existing resources of human-generated concreteness norms [60], and an analysis of the directed responses, i.e., responses in which one candidate is directly referring to the other. The first analysis looks at the complexity of the candidate's responses. If one candidate uses more abstract words than the other, this indicates that they are more complex and less direct. The second method looks at directness from the perspective of grounding their response as a direct attack or argument against the opponent.

**Word concreteness analysis** provides the first part, for which we aimed to quantify the complexity of language used by each candidate to evaluate word concreteness. Concreteness scores reflect how tangible or abstract a word is, with higher scores indicating more concrete, easily visualisable terms, and lower scores representing more abstract, complex terms. Brysbaert et al.'s [60] word concreteness norms were utilised, as they provide concreteness ratings for approximately 40,000 words.

The text data from the debate was preprocessed before analysis. This preprocessing involved lemmatisation to standardize word forms and the removal of stop words to eliminate high-frequency but low-information terms. For each remaining word, we retrieved its concreteness score from Brysbaert's list (https://github.com/ArtsEngine/concreteness/blob/master/Concreteness_ratings_Brysbaert_et_al_BRM.txt). We then computed the average concreteness score for each candidate's speech to assess whether one candidate used more concrete (direct) or abstract (complex) language on average.

To determine if there were statistically significant differences in concreteness scores between the candidates, we first conducted a Shapiro-Wilk test for normality. Based on the normality results, we applied either a parametric test or a non-parametric alternative (Mann-Whitney U test). Additionally, we performed a qualitative analysis by selecting and comparing the five most concrete and five most abstract words used by both candidates, offering insights into the kinds of terms each favoured.

**Direct reference** of the opponent is analysed separately for each candidate. For Trump, we count both direct mentions of *Kamala* and *Harris* and indirect mentions such as *she*, *her*, and *hers*. Similarly, for Harris, we count direct mentions of *Donald* and *Trump*, as well as indirect references like *he*, *him*, *his*, and *he's*.

For indirect mentions, we manually resolve co-references to ensure that *he* refers to Trump or *she* refers to Harris when necessary. While Fig 3 already shows a high frequency of Harris mentioning *Donald Trump*, this analysis provides a more focused look at direct attacks. Additionally, since *Biden* is the most frequently mentioned entity in Trump's responses, we also track the mentions of *Joe* and *Biden*.

## 6.2 Results

**Word concreteness analysis** revealed a slight difference in the average concreteness scores between Trump and Harris. Trump's responses had an average concreteness score of 2.03 (SD = 1.38), while Harris' responses had a higher average concreteness score of 2.10 (SD = 1.31). Although both candidates used relatively abstract language overall (as indicated by scores closer to the lower end of the concreteness scale), Harris' language was marginally more concrete. The standard deviation values indicate that Trump's language exhibited slightly more variability in concreteness compared to Harris'.

The non-normality of both Trump's and Harris' concreteness scores, as confirmed by the Shapiro-Wilk test (Trump: SW = 0.826, p < 0.001; Harris: W=0.862, p < 0.001), justified the use of the non-parametric Mann-Whitney U test for comparing the two candidates. The Mann-Whitney U test revealed a statistically significant difference in concreteness between the two (U = 0.002, p < 0.01), suggesting that despite the overall abstract nature of both candidates' language, Harris' responses were marginally more concrete than Trump's.

The qualitative analysis of the most concrete and abstract words used by Trump and Harris further highlights differences in their linguistic styles. In Trump's responses, the most concrete words included terms like *people* (76 occurrences), *person* (8 occurrences), and *student* (7 occurrences), indicating a focus on specific, tangible entities. Conversely, his most abstract words were *go* (110 occurrences), *get* (80 occurrences), and *country* (68 occurrences).

Similarly, in Harris' responses, concrete words such as *people* (49 occurrences), *bill* (6 occurrences), and *senator* (3 occurrences) pointed to a focus on identifiable subjects, though with fewer mentions compared to Trump. Harris' most abstract terms included *Trump* (38 occurrences), *know* (33 occurrences), and *would* (32 occurrences), suggesting a greater emphasis on referencing her opponent and hypothetical or uncertain situations.

Overall, both candidates used a mix of concrete and abstract language, with notable differences in the types of words they prioritised. Trump relied more heavily on action-oriented abstract terms like *go* and *get*, while Harris focused more on names and references to her opponent, such as *Trump* and *Donald*, which leads over to the analysis of direct references.

**Direct references** analysis reveals distinct differences in how Trump and Harris referred to one another during the debate. Notably, Trump made no direct reference to Harris by name throughout his responses. In contrast, Harris directly mentioned Trump or Donald 70 times, showcasing a much more pointed focus on the name of her opponent. In contrast, Trump's references to Biden (or "Joe") were 15 direct mentions.

When considering how often each candidate referred to their opponent by name within their overall responses, Trump did not refer to Harris at all. On the other hand, Harris referred to Trump or Donald by name in 19 responses, amounting to 55.88% of her responses.

The indirect mentions further highlight these patterns. Trump indirectly referred to Harris in 34 responses, which is 45.95% of his total responses. Harris, in comparison, indirectly referred to Trump in 19 responses, again 55.88% of her responses. It is worth noting that we excluded three instances in which Harris referred to either Zelenskyy or Putin with the pronouns *he* or *him*, accounting for the adjusted count of 19 indirect references to Trump. Considering both, direct or indirect reference, Trump referred to Harris in 34 responses, which is 45.95% of his responses. Harris referred to Trump in 22 responses, which is 64.71% of her responses (see Fig 7 for an overview).

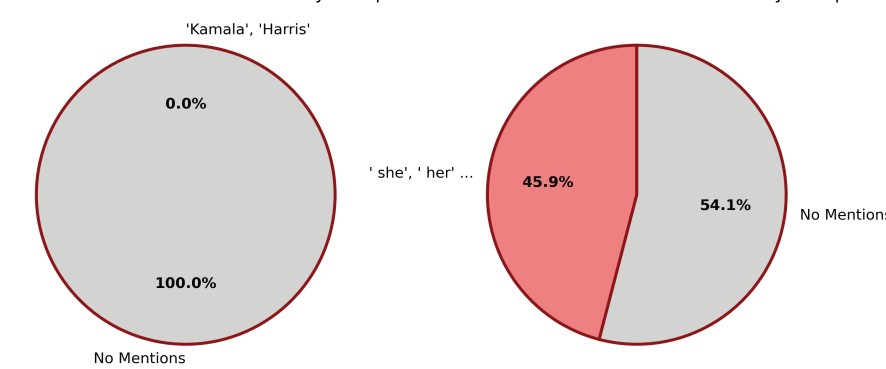

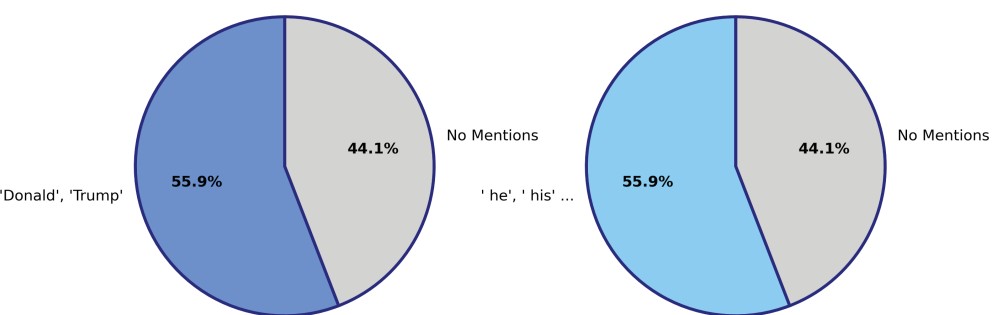

**Fig 7. Charts that visualise the amount of references that could be identified across all responses.** Upper charts (red border) show the direct (left) and indirect (right) mentions of Kamala Harris by Donald Trump. Trump never mentions "Kamala" or "Harris" directly. The lower charts (blue border) show the direct and indirect mentions of Donald Trump by Harris.

# 7 Study 5: identity politics and group appeal

## 7.1 Method

**Identity Politics** are assessed in this study by examining whether each candidate's responses align with the core values of their respective political parties. For Trump, we focus on the ideals of the Republican Party, while for Harris, we consider the values of the Democratic Party. It is important to note that Trump's foreign and security policies, as the 45th U.S. President, often diverged from traditional Republican positions [61].

To quantify the extent to which a candidate's response reflects party ideologies, we utilise the Political DEBATE (DeBERTa) Model (https://huggingface.co/mlburnham/Political_DEBATE_base_v1.0). This model is based on DeBERTa (Decoding-enhanced BERT with disentangled attention), a cutting-edge approach to Natural Language Inference (NLI), specifically fine-tuned for classifying political texts [12]. The model excels in both zero-shot and few-shot learning tasks, making it ideal for the classification of political discourse.

In our analysis, we employed the model to test two hypotheses for each candidate's debate responses:

- $H_1$: This text expresses Republican beliefs.
- $H_2$: This text expresses Democrat beliefs.

The model returns a probability score for each hypothesis, indicating the likelihood that the given text reflects the specified political ideology. We applied this method separately to Trump's and Harris' responses, measuring the extent to which each aligns with either Republicans' or Democrats' principles. Each response was classified into one of four categories based on the model's output:

- Neither party ($H_1$ false, $H_2$ false)
- Republican ideals ($H_1$ true, $H_2$ false)
- Democrat ideals ($H_1$ false, $H_2$ true)
- Both parties ($H_1$ true, $H_2$ true)

Responses classified as aligning with neither or both ideologies were excluded from further interpretation. The model assigns a probability to each response ($i$ responses) with probabilities $p_i$ close to 0 indicating that the hypothesis is *not* rejected. Hence, for our interpretation, we take $1 - p_i$ as the probability for the response to be likely expressing republican or democrat ideals. We only count those responses to belong to a party ideal if the probability for rejection is $p_i < 0.05$.

**Group Appeal** is the second part of the analysis, a concept that reflects how candidates position themselves in relation to the broader electorate and the social identities of their supporters. As outlined in the Theoretical Background, we hypothesise that the language used by Harris will demonstrate a stronger focus on inclusivity and collective identity, as evidenced by a higher frequency of inclusive pronouns such as "we" and "us." This linguistic strategy is often employed to evoke a sense of unity, shared purpose, and solidarity, particularly among voters who identify with the Democratic Party's ideals of social equity and communal action. Harris' use of such pronouns would thus be indicative of an appeal to collective responsibility and a desire to foster a sense of belonging within a broader societal context.

In contrast, we hypothesise that Trump's rhetoric will show a greater emphasis on individualism and personal authority, highlighted by a more frequent use of pronouns such as "I" and "me". This is consistent with his well-documented communication style, which often centres around his own leadership, decisions, and personal successes.

To systematically assess these tendencies, we analyse the frequency of both plural pronouns (e.g., "we", "us") and singular pronouns (e.g., "I", "me") across all responses given by Trump and Harris during the debate.

## 7.2 Results

**Political Affiliation** analysis results are presented in Fig 8. The analysis categorised the candidates' debate responses based on whether they expressed Republican or Democrat beliefs. For Trump, 10 responses (13.70%) were classified as Republican, while 4 of his responses (5.48%) were categorised as Democrat. In contrast, for Harris, 2 responses (5.88%) were classified as Republican, while 6 responses (17.65%) were identified as Democrat.

To further investigate the results, we examined the specific instances where a candidate's response was classified as aligning with the opposing party's ideology. In four of Trump's responses, the Political DEBATE model labelled his statements as Democratic, while two of Harris' responses were labelled as Republican. Trump's responses classified as Democratic include references to policies traditionally associated with Democratic ideals, such as critiques of foreign trade, immigration, and domestic policy. For instance, Trump criticises Harris' position on various domestic issues, but in doing so, discusses concepts like social justice, immigration policy, and economic concerns in ways that overlap with Democratic rhetoric

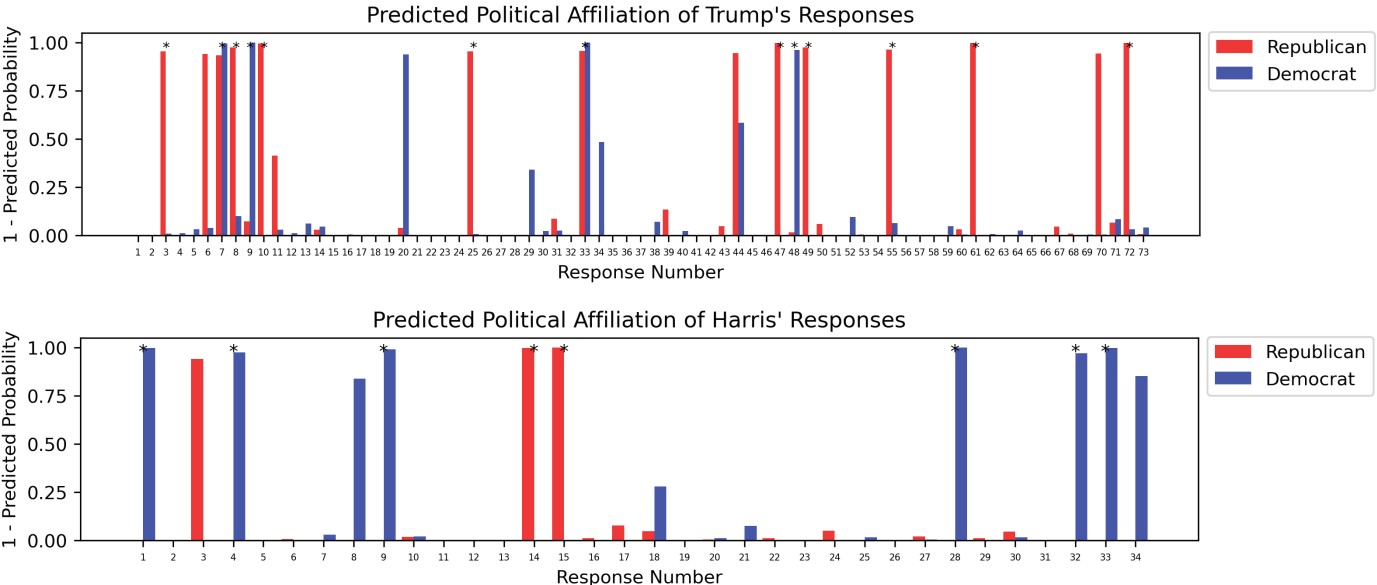

**Fig 8. Probabilities of each response by Trump (top) and Harris (bottom) with respect to the hypothesis that their response expresses republican beliefs (red) or democrat beliefs (blue).** The NLI model rejects the hypothesis with the predicted probability, hence, the y-axis shows the 1 - predicted probability, the probability that the response expresses a party belief. * indicates that the probability of the model to not reject the hypothesis is $p_i < 0.05$.

(e.g., his references to criminal justice reform and social policy, despite positioning them in opposition to his views). Additionally, his remarks about renewable energy and climate policy, while critical, touch on themes more commonly aligned with Democratic discourse.

Similarly, Harris' two responses labelled as Republican also reflect a degree of crossover. In one response, she emphasises her track record of prosecuting transnational criminal organisations and supporting increased border security measures, which are typically Republican stances on law and order. Additionally, she highlights her bipartisan support from former Republican figures, including endorsements from prominent Republican members, which may have contributed to the Republican classification by the model.

These responses highlight the complexity of political affiliation classification in rhetoric, as elements of both parties' ideologies may appear within a broader critique, as well as the fact that political discourse in debates is not always strictly aligned with party ideology. There are moments where candidates address issues in ways that might resonate across the political spectrum, particularly when discussing topics like criminal justice, border security, or economic policies.

**Group Appeal** analysis reveals distinct differences in the pronoun usage between Trump and Harris, highlighting their contrasting rhetorical styles (see Fig 9).

Trump's responses in the debate were heavily skewed toward individualistic language. He used first-person singular pronouns ("I," "me," "my," "mine") 245 times, in contrast to 133 instances of inclusive pronouns ("we," "us," "our," "ours"). This indicates that Trump employed 84.21% more singular pronouns than plural ones, underscoring his rhetorical focus on personal authority and leadership. This aligns with his historical communication style, emphasising self-reliance and decisive leadership.

Harris, on the other hand, demonstrated a more balanced use of both pronoun types. While she used first-person singular pronouns 144 times, her use of inclusive pronouns totalled 136. The difference, at 5.88%, suggests that Harris employed a near-equitable blend of

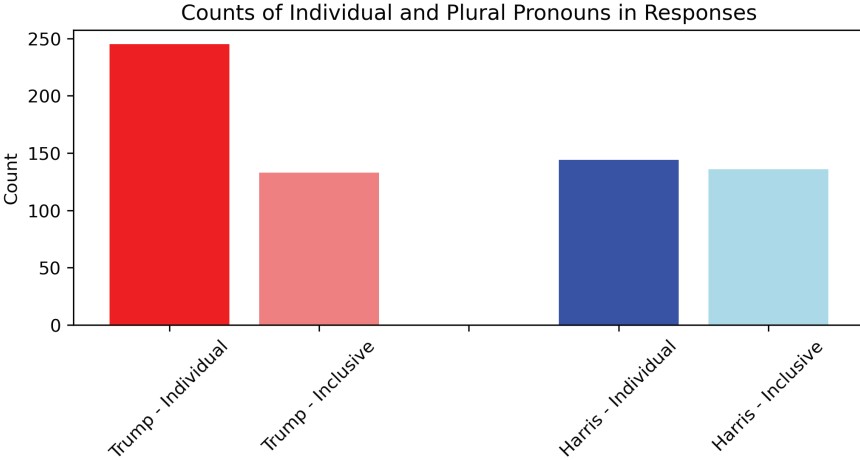

**Fig 9. Visualisation of the counted pronouns relating to the individual ("I," "me," "my," "mine": 245 times for Trump in red) and those that are inclusive ("we," "us," "our," "ours": 133 times for Trump in light red). For Harris, the blue bar indicates individual ("I," "me," "my," "mine": 144 times) counts of pronouns and inclusive count ("we," "us," "our," "ours": 136 times in light blue).**

individualistic and collective language. This supports our hypothesis that her rhetoric would more strongly reflect inclusivity and collective identity, consistent with her positioning within the Democratic Party.

## 8 General discussion and conclusion

This study provides a linguistic analysis of the debate held on September 10th, 2024 between Trump and Harris, focusing on the candidates' alignment with the broader communicative strategies that typically characterise their respective parties. By analysing the language used by both candidates through a combination of quantitative and qualitative methods, we aimed to uncover how each candidate's rhetoric reflects the stereotypical linguistic patterns associated with Republican and Democratic ideologies. The following discussion synthesises the key results from each section, and their interpretation based on the initial hypotheses formulated in the Theoretical Background section.

Overall, a preliminary comparison of the most frequently used words by the two candidates reveals that Trump's top words ('like', 'vote', 'thing', 'bad', and 'never') are characteristic of spoken discourse and suggest a deliberate choice to project relatability and spontaneity, while also conveying negative emotion and action-oriented rhetoric. In contrast, Harris' top five distinctive words ('Donald', 'American', 'understand', 'work', and 'States') indicate an intention to personalize her critiques of the opponent, engage directly and assertively, and emphasize themes of national unity and commitment to public service.

More specifically, in our **first study** in Section 3 we explored the use of **figurative frames** in the two candidates' language. To achieve that, we used a combination of automatic identifications of figurative frames, and manual annotations on the identified frames, followed by interrater agreement tests and analyses over the finalised annotations. Overall, we found that Harris used substantially more statements that evoke figurative frames, compared to Trump. Regarding the type of frames and figurative constructions used by the two candidates, we

observed some differences in how Trump and Harris frame issues of national concern, reflecting their broader political ideologies and communication styles. These differences are consistent with established patterns in Republican and Democratic rhetoric, with Trump focusing on crisis and decline, and Harris on recovery and empowerment. As further discussed in the final paragraphs of this section, the methodological innovation of this study is that of using state-of-the-art computational tools to assist in the identification and classification of frames (GPT-4o model) combined with human annotation, to validate the results. The automatic method proved to be reliable, therefore opening new research pathways for the identification of figurative frames in political discourse. Future research may focus on quantifying the specific metaphor-related words used by the candidates at a lexical level, using fine-grained procedures like MIPVU ([62] or DMIP [63]) which at the moment are fully manual but may be integrated in more complex methodological architectures that involve Large Language Models.

In our **second study** in Section 4, we focused on the **emotional valence** (polarity) of the words used by the candidates. We hypothesised that Trump's words would be characterised by a stronger negative emotional valence, while Harris' rhetoric would reflect a positive emotional tone. Contrary to our initial hypothesis, we found that on average the words used by the two candidates do not differ significantly in emotional valence, even though there is a tendency of Trump's words to be slightly more negatively valenced, compared to Harris. Overall, both rely on negative sentiment to a considerable extent. This is in line with some recent findings by Maier and Nai (2020) [64], in which the authors report a cross-national study of 107 national elections, and argue that emotional appeals, particularly negative emotions such as fear, anger, and disgust, play a central role in modern political communication. Negative emotions, they contend, are shown to be more effective than positive ones in securing media coverage and mobilizing voters, as they tend to heighten perceptions of threat, urgency, and moral outrage. Rather than being detrimental, negativity is often a strategic tool used to define opponents, reinforce group identity, and drive engagement in competitive electoral contexts. Moreover, the results from the statistical analyses in our study indicate that the sentiment scores for both candidates do not follow a normal distribution, suggesting that their use of positive and negative sentiment may vary considerably depending on the topic of discussion, throughout the debate. The analysis of the subjectivity scores suggests that there is a significant difference in the average subjectivity of the words used by Trump and Harris during the debate, with Harris using words with higher average scores of subjectivity. We interpret this finding, suggesting that Harris' rhetoric includes a greater explicit mention of her personal opinions and views on public issues. This is consistent with her frequent appeals to empathy and moral judgment, as seen in the qualitative analysis of her most positive and negative responses, where she often frames issues in terms of human impact and ethical considerations. In contrast, Trump's subjectivity scores generally suggest a more objective tone, where the candidate seeks to present what he views as facts. The non-normal distribution of subjectivity scores for both candidates, confirmed by the statistical test, also suggests that their use of subjective language varies widely, depending on the specific topics being addressed.

In the **third study** in Section 5, we assessed the levels of lexical specificity of responses comparing the words used by Trump and Harris. We argued that this analysis would provide insights into how these political figures present policy-related information. By quantifying the specificity of content words in their responses, we aimed to distinguish between detailed, specific lexical choices and broader, big-picture language use. Despite the documented differences in rhetorical styles between political parties, our results indicate that there is no significant difference in lexical specificity between the two candidates' responses. This finding is noteworthy because it contrasts with the expectation that Harris, often associated with more

policy-dense discussions, would use more specific language, whereas Trump might rely on less specialised terminology, in order to engage with a wider audience. A possible explanation could be that the candidates were prompted to discuss the same issues, in the same context, addressing the same audience.

Another factor to consider is the method used to quantify specificity. While the use of WordNet's depth hierarchy allows for a systematic analysis, it might not fully capture nuances such as the precision of policy explanations. Emerging research by the Abstraction group has begun to address these limitations by developing human-annotated specificity ratings using Best-Worst Scaling (BWS) methods ([65,66]) and scaling up the datasets using LLMs [67]. These ratings provide a more empirically grounded and cognitively plausible understanding of lexical specificity, and given the reliability and higher flexibility obtained with their approximation through LLMs, this method may be used in the near future to look deeper into the role of this lexical variable in political discourse. Furthermore, since our analysis focused on individual words, it may have overlooked sentence structure or broader rhetorical strategies that contribute to perceived specificity. It is also important to note that the number of responses analysed differed between the two candidates, with Trump providing more responses than Harris. Although our methodology controls for response length and ensures that average specificity is comparable, this imbalance in the dataset could potentially mask finer distinctions in the depth of policy discussion between the two.

In the **fourth study** in Section 6, we tackled the semantic complexity of the lexicon employed by the two candidates, comparing the average concreteness of the words they used in their responses. We argue that the more concrete the words, the clearer the topic is addressed. In contrast, the more abstract the words, the more difficult it is to process the response. This assumption is based on extensive previous literature on the concreteness effect, showing that on average concrete words are easier to process, compared with abstract words (e.g., a recent literature review in [68]). It is also supported by extensive empirical research in metaphor studies, which shows that speakers often use metaphorical language to render abstract concepts more concrete, thereby enhancing clarity and comprehensibility when communicating complex ideas [69]. We hypothesised that, based on previous literature, Trump would use more concrete words to reach a wider audience, while Harris would use more abstract words, to appeal to an educated and progressive audience. We found that both candidates used predominantly abstract language, as indicated by their scores leaning toward the lower end of the concreteness scale. This pattern may be understood in light of recent findings suggesting that abstract language in spoken discourse tends to elicit greater curiosity and engagement from listeners, whereas concrete language is more often associated with a sense of closure and reduced motivation to further engage with the speaker [70]. Moreover, we find a significant difference between the average scores of the words used by the two candidates that goes in the opposite direction of what we hypothesised: Harris uses more concrete words compared to Trump. We interpret this as an intentional effort by Harris to adopt a direct and concrete language (typically a characteristic of Republican language), to reach a wider audience, given the context of the debate and the need to persuade undecided voters. Additionally, we found that Trump made no direct mention of Harris by name, while Harris referred to Trump or "Donald" 70 times. This may be due to rhetorical choices as well as practical dynamics such as the fact that Harris' first name was previously mispronounced by the Republican candidate and this phenomenon had become central in various critical judgments of his campaign. Another reason could be that Trump's preparations and strategic focus for the debate may have centred more on addressing Joe Biden's policy and governance, rather than Harris'. These arguments could explain his decision to refrain from mentioning Harris directly and by name, throughout his responses.

In our **fifth study** in Section 7, we found that overall both, Trump's and Harris' statements align with the typical language used by their respective parties. Moreover, looking at the use of pronouns (singular and plural) we found that Harris exhibited a nearly equal use of plural pronouns such as "we" and "us" and singular pronouns like "I" and "me," underscoring her commitment to inclusivity and collective identity, which aligns with her Democratic positioning. Trump, conversely, used significantly more singular pronouns, reinforcing his personal leadership style and decisiveness. Stepping back to observe broader patterns in our data, we note that certain tendencies identified in individual studies appear to converge in meaningful ways. For instance, Trump's frequent use of self-referential language (naming himself among the top 5 named entries) is consistent with the data from the Group Appeal analysis where Trump used more individualistic language (Study 5). This suggests that linguistic features such as pronoun use may not operate in isolation but instead reflect deeper rhetorical and ideological structures.

Finally, the analysis also reveals a few cases in which both candidates adopted rhetoric typically associated with the opposing party. Trump's references to social justice and climate policy, albeit critical, showed overlap with Democratic themes, while Harris highlighted traditionally Republican stances like border security. First of all, this phenomenon suggests a possible limitation of the approach, which may fall short in detecting the real communicative intentions of the speaker when debating a topic, and secondly the idea that political debate discourse is arguably more fluid and linguistically balanced, compared to rallies, with candidates occasionally appealing to broader audiences by incorporating elements from the opposing party's ideology.

Overall, our findings provide a linguistic perspective on the alignment between the speech profiles of Trump and Harris and the lexical and semantic characteristics typically associated with Republicans and Democrats. While several measures, such as pronoun use and sentiment markers, conformed to established partisan patterns, others deviated, suggesting that contextual factors and individual rhetorical styles also play a significant role. Rather than weighing any single measure more heavily, we interpret the collective patterns across multiple dimensions to assess the broader alignment with partisan speech norms. Therefore, our findings show that while some rhetorical patterns align predictably with partisan identities, others reflect calculated shifts aimed at reaching broader audiences. Crucially, the study also offers a methodological advancement: a scalable, mixed-method approach to analysing figurative frames that blends the analytical power of large language models with the interpretive depth of manual annotation. This hybrid method not only enriches our understanding of political discourse but sets a precedent for future research. The central takeaway is clear: language in political debates is neither fixed nor purely partisan: it is dynamic, context-sensitive, and deeply revealing of both strategy and ideology.

In conclusion, political debates serve as distinct arenas of language use where candidates must appeal to a broad audience, unlike rallies, which focus on energizing their base. During rallies, candidates reinforce established views and mobilize loyal voters, whereas in debates, they must confront their opponent directly and strategically appeal to undecided voters or even sway those from the opposing side. This necessity for broader appeal may explain why we observed relatively few significant linguistic differences between Trump and Harris: both candidates likely adapted their rhetoric to resonate with a wider electorate. However, the differences that did emerge are particularly telling, as they reflect the strategic choices each candidate made in navigating the high-stakes debate format, which demands a more calculated approach to audience engagement compared to the unfiltered messaging of rallies.

A final note. When we first wrote this manuscript, the outcome of the U.S. election was still unknown. Now, with Trump's victory confirmed, the implications of political discourse feel

even more pressing. The way candidates communicate not only shapes electoral outcomes but also influences broader political and geopolitical landscapes. If one thing remains clear, it is that words and their use continue to hold immense power.

## Usage of AI Tools and Technologies

In line with the PLOS ONE Ethical Publishing Practice (https://journals.plos.org/plosone/s/ethical-publishing-practice#loc-artificial-intelligence-tools-and-technologies) we hereby report any artificial intelligence (AI) tools and technologies.

- *OpenAI - ChatGPT (Model GPT-4o)*: Rephrasing, formulating, formatting and correcting written content and code presented in this paper.
- *OpenAI - API (Model GPT-4o)*: Generation and interpretation of the figurative frames in Study 1 (Sec. 3.1). Formatting of bibtex entries, proofreading of parts of the text.
- *GitHub Copilot (Model GPT-3.5 for code completions)*: Coding assistance, code commenting and code formatting.

We declare that, to the best of our knowledge and with revision of both authors, the content is accurate and valid, there are no concerns about potential plagiarism, all relevant sources are cited, all statements in the article reporting hypotheses, results, conclusions, limitations, and implications of the study represent the authors' own ideas - with exception of the interpretation of the figurative framings (details on verification of these results are reported in Sec. 3.1).

## Author contributions

**Conceptualization:** Philipp Wicke, Marianna M. Bolognesi.

**Data curation:** Philipp Wicke, Marianna M. Bolognesi.

**Formal analysis:** Philipp Wicke, Marianna M. Bolognesi.

**Funding acquisition:** Marianna M. Bolognesi.

**Investigation:** Philipp Wicke, Marianna M. Bolognesi.

**Methodology:** Philipp Wicke, Marianna M. Bolognesi.

**Project administration:** Philipp Wicke, Marianna M. Bolognesi.

**Resources:** Philipp Wicke, Marianna M. Bolognesi.

**Software:** Philipp Wicke.

**Validation:** Philipp Wicke, Marianna M. Bolognesi.

**Visualization:** Philipp Wicke.

**Writing – original draft:** Philipp Wicke, Marianna M. Bolognesi.

**Writing – review & editing:** Philipp Wicke, Marianna M. Bolognesi.

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
