## [Decision Letter · Decision Letter 0]

5 Jan 2025

PONE-D-24-47133Red and blue language: Word choices in the Trump & Harris 2024 presidential debatePLOS ONE

Dear Dr. Bolognesi,

Thank you for submitting your manuscript to PLOS ONE. After careful consideration, we feel that it has merit but does not fully meet PLOS ONE’s publication criteria as it currently stands. Therefore, we invite you to submit a revised version of the manuscript that addresses the points raised during the review process.

We look forward to receiving your revised manuscript.

Kind regards,

Michal Ptaszynski, PhD

Academic Editor

PLOS ONE

Journal Requirements:

“MB is funded by the European Research Council (ERC-2021-STG-101039777, project ABSTRACTION). Views and opinions expressed in the present paper are however those of the author(s) only and do not necessarily reflect those of the European Union or the European Research Council Executive Agency. Neither the European Union nor the granting authority can be held responsible for them.”

“Marianna Bolognesi is partly funded by the European Research Council 913 (ERC-2021-STG-101039777, project ABSTRACTION). Views and opinions expressed in the present paper are however those of the author(s) only and do not necessarily reflect those of the European Union or the European Research Council Executive Agency. Neither the European Union nor the granting authority can be held responsible for them.”

“MB is funded by the European Research Council (ERC-2021-STG-101039777, project ABSTRACTION). Views and opinions expressed in the present paper are however those of the author(s) only and do not necessarily reflect those of the European Union or the European Research Council Executive Agency. Neither the European Union nor the granting authority can be held responsible for them.”

Reviewers' comments:

Reviewer's Responses to Questions

**Comments to the Author**

1. Is the manuscript technically sound, and do the data support the conclusions?

Reviewer #1: Partly

Reviewer #2: Yes

2. Has the statistical analysis been performed appropriately and rigorously? 

Reviewer #1: Yes

Reviewer #2: Yes

3. Have the authors made all data underlying the findings in their manuscript fully available?

Reviewer #1: Yes

Reviewer #2: Yes

4. Is the manuscript presented in an intelligible fashion and written in standard English?

Reviewer #1: Yes

Reviewer #2: Yes

5. Review Comments to the Author

Reviewer #1: I like the general idea of the contribution. The corpus seems rather modest for the amount of analyses carried out, but the manuscript can be considered a case study that shows how such analyses can be done.

General remarks, suggestions:

One of the most innovative aspects of the study is the use of the OpenAI LLM to identify the framing. While the inter-rater agreement of the human raters is discussed, it would be particularly relevant to know to what extent there is consistency in what the LLM does and the ratings. I understand that the raters worked on the framings identified by the LLM only (and did not themselves work through the corpus to identify the framings). It would still be interesting to get an idea of how many instances were considered ‘falsely identified’ by the raters.

Another general comment would be that while the number of indices generated with automatic processing is impressive, it is not always very clear why these indices (termed “studies”) are always relevant for the analysis of political discourse.

Even though in general I genuinely think that the case study is potentially interesting for scholars working on political discourse, I believe the authors should do a more thorough job explaining why the specific operationalizations are adequate (and not simply feasible because there are tools on huggingface that can do something).

Thus I recommend major revisions.

Some details:

l. 396� “Recognizing the possibility of sentence dependence within responses, we acknowledge that some interdependence may exist between sentences, but argue that this dependence does not diminish the value of the statistical test, as political rhetoric often varies greatly sentence by sentence, making sentence-level analysis appropriate for capturing these shifts.”

The last statement is not supported by any citations or examples. I find it difficult to understand what exactly is meant, it looks a bit as if the problem of lack of independence of the sentences is explained away rather superficially. Please provide some more substantial argument here.

559 � “By maintaining a balance between presenting factual information and personal reflections, Harris’ objective responses are grounded in verifiable claims, while the subjective ones include more emotive, value-laden statements.”

This concluding statement is rather trivial and tautological: it is not a result but the author’s not really surprising assumption and operationalization that subjective statements are more value-laden.

565 Not everybody know the construct of depth in the lexical hierarchy as it is implemented in WordNet. Please give some explanation what this refers to, and some examples. Is it related to lexical sophistication (which would be a more established construct in vocabulary research).

In this context, the authors argue that they focus on content words as an operationalization of Details vs. Big Picture and “filter out function words such as prepositions and conjunctions. This decision ensures that the analysis emphasises semantically rich vocabulary, thereby enhancing the robustness of our findings.”

This is not very convincing, as complexification also via conjunctions arguably allows speakers to provide a more nuance discussion than simply chaining main clauses. One has the impression that the feasibility and ease of obtaining scores from automatic algorithms trumps (pun intended) a thorough operationalization of the target constructs.

The conclusion (602) that both candidates show comparable levels of **lexical** specificity is certainly empirically correct, but it should not be confounded with complexity in terms of their statements, for the aforementioned reasons.

614 � Complexity of the language is operationalized via word concreteness. No rationale is given for this rather surprising choice, which is problematic given the huge amount of literature on complexity in first and second language production (cf. e.g. work by Biber, Housen, etc. etc.). Moreover, I do not understand why directness and complexity are treated as trade-offs, they seem rather orthogonal. Please explain why this should be the case, and provide relevant citations.

763 � “Similarly, Harris’ two responses labelled as Republican also reflect a degree of crossover.” Again, this is a tautological ‘conclusion’: The very operationalization of the authors is that whenever a statement fits in with the opposite party’s ideology, it is by definition crossover.

859 � “We argued that the more concrete the words are, the clearer the topic is addressed.”

I come back to the problematic use of lexico-semantic concreteness: In the conclusions, the authors now associate concreteness with clarity, while in the actual study concreteness was a sign of simplicity. Simplicity and clarity are not the same, an utterance may be complex but still clear; if the content matter is complex and abstract, it may be impossible to make it concrete while it may still be possible to remain clear. The analytical dimension complexity, clarity, and concreteness need to be better defined and more solid rationales need to be provided why and how they are related.

Reviewer #2: This manuscript compared semantic and pragmatic aspects of the language used by Kamala Harris and Donald Trump during the presidential debates. The specific measures of interest were: (a) the use of figurative frames, (b) the emotional valence (polarity) of produced words, (c) lexical specificity, (d) word concreteness, and (e) use of pronouns. The results revealed some differences between the two candidates, i.e., Harris used more statements that evoked figurative frames and more concrete words than Trump, where Trump used proportionally more singular pronouns than Harris. In contrast, there were no differences in use of emotional words or in lexical specificity. These findings are discussed in terms of the alignment of the candidates’ communicative strategies with differing political ideologies.

I enjoyed reading this manuscript, which is well-written and has clearly stated hypotheses to be tested. The hypotheses are well-grounded in previous literature underlying language use in speakers of different political parties. However, I do have some concerns related to the narrowness of the paper’s scope and the lack of synthesis across the measures. Specific comments are given below.

Theoretical Development and Literature Review:

- The previous research that is cited is overly specific to differences in speech between political parties. While that is necessary, the paper would benefit from connecting more broadly to literature beyond political speech. For example, there is a wealth of literature on psychosocial constructs that underlie word choices (e.g., see James Pennebaker’s work) or on linguistic relativity (e.g., see Lera Boroditsky’s work; note also a paper by Chkhaidze et al., 2021, that specifically tests the influence of linguistic metaphors on attitudes toward immigration).

- While the authors do a good job of reviewing relevant literature for each aspect, what is missing for me is cohesion in bringing these particular measures together. Why were these particular measures selected, of the many that could have been analyzed? Is there any theory that ties them together?

- Another reference to consider for broader perspective on natural language processing: Berger, J., & Packard, G. (2022). Using natural language processing to understand people and culture. American Psychologist, 77(4), 525–537. https://doi.org/10.1037/amp0000882

Results and Analyses:

- Again, in thinking about broader applicability, it would be interesting to compare measures in the present paper using Linguistic Inquiry and Word Count (LIWC-22) software, which counts words in psychologically meaningful categories. While not as sophisticated as the algorithms and models used to analyze the results in this paper, LIWC has been extensively used for text analysis in many papers and therefore could be useful in comparing whether some of the present results do or do not replicate. LIWC has measures of pronouns, positive and negative emotion words, and positive and negative tone, which could be comparable to similar measures reported in this paper. LIWC also includes measures of other psychological processes that could assess some of the hypothesized differences between Harris and Trump (e.g., drives such as achievement and power, or motives such as reward and risk).

- There is a lot to keep track of in this paper, so it would be helpful to have a table that shows each measure, the hypothesis for it, and the result.

- The authors should think about which figures are most critical for presenting the results (which ones are necessary to display the data from testing each of the hypotheses) and whether a subset of the data would be better presented in tables or in text.

Discussion:

- While the manuscript interprets each study’s results independently, I didn’t have a clear sense of the cumulative conclusions with respect to the central research question: “Are speech profiles of Trump and Harris systematically aligned with the typical characteristics identified in the literature as peculiar to Democrats and Republicans?” Not all of the measures demonstrated the expected patterns, so what is the overall conclusion? Are there measures that are given more weight than others when assessing the answer to this question?

- What is the contribution of this research beyond describing the speech profiles of two people? Some discussion of how the results extend previous research and tell us something new would be useful in documenting the research’s contribution to the literature.

6. PLOS authors have the option to publish the peer review history of their article (what does this mean?). If published, this will include your full peer review and any attached files.

Reviewer #1: No

Reviewer #2: No

---

## [Author Response · Author response to Decision Letter 1]

18 Feb 2025

Thank you. Please find the information in the Reply to reviewers and the updated financial statement in the Cover letter, as requested.

---

## [Decision Letter · Decision Letter 1]

3 Apr 2025

PONE-D-24-47133R1Red and blue language: Word choices in the Trump & Harris 2024 presidential debatePLOS ONE

Dear Dr. Bolognesi,

Thank you for submitting your manuscript to PLOS ONE. After careful consideration, we feel that it has merit but does not fully meet PLOS ONE’s publication criteria as it currently stands. Therefore, we invite you to submit a revised version of the manuscript that addresses the points raised during the review process.

We look forward to receiving your revised manuscript.

Kind regards,

Michal Ptaszynski, PhD

Academic Editor

PLOS ONE

Journal Requirements:

**Additional Editor Comments:**

Reviewer #2:

I am very appreciative of the authors’ receptiveness to my constructive suggestions as well as their thoughtful efforts to address the reviewers’ major points in their cover letter and in this revision. The manuscript has improved in several respects, in particular a broader literature reviewer and a clearer connection between the analyses and the hypotheses they are trying to address.

My remaining issue is regarding the manuscript’s general discussion and conclusion section, which could benefit from more connection back to the literature raised in the introduction and theoretical background sections (there are virtually no citations in the discussion). It would be helpful for the reader to see a better integration between these sections and also help to ground some of the conclusions in previous research.

Relatedly, I would like the authors to more explicitly address limitations of their research as well as directions for future research that build on their findings. They give a couple of examples in specific studies (one limitation in the fifth study and one future research direction in the first study), but a broader perspective on these issues would help to address my previous review comment about missing some cohesion in bringing these measures together. I understand that no overarching theoretical framework exists, but the authors could find other ways to connect the different studies, such as a limitations/future research section. I also see ways that the data in different studies could be linked; for example, Trump naming himself among the top 5 named entries (Study 1, Figure 3) seems consistent with the data from the Group Appeal analysis where Trump used more individualistic language (Study 5). Having the authors reflect on findings across studies, wherever possible, would enhance the reader’s understanding of the findings and their implications.

Reviewers' comments:

Reviewer's Responses to Questions

**Comments to the Author**

1. If the authors have adequately addressed your comments raised in a previous round of review and you feel that this manuscript is now acceptable for publication, you may indicate that here to bypass the “Comments to the Author” section, enter your conflict of interest statement in the “Confidential to Editor” section, and submit your "Accept" recommendation.

Reviewer #1: All comments have been addressed

Reviewer #2: (No Response)

2. Is the manuscript technically sound, and do the data support the conclusions?

Reviewer #1: Yes

Reviewer #2: Yes

3. Has the statistical analysis been performed appropriately and rigorously? 

Reviewer #1: Yes

Reviewer #2: Yes

4. Have the authors made all data underlying the findings in their manuscript fully available?

Reviewer #1: Yes

Reviewer #2: Yes

5. Is the manuscript presented in an intelligible fashion and written in standard English?

Reviewer #1: Yes

Reviewer #2: Yes

6. Review Comments to the Author

Reviewer #1: I would like to thank the authors for the revised text. It is now more accessible for readers who might not be familiar with some or all of the NLP tools used. I genuinely think this is a highly useful contribution to the field as the methodologies described can be used for other corpora and to some extent also for other languages.

Minor comments that I have after reading the paper once again thoroughly:

Is it really necessary to aggregate the scores, e.g. the sentiment labels, for whole turns in the debate? I see why you want to do it to understand the timeline, but wouldn't it be better to treat each sentence as a unit of analysis and fit a logistic model with the predictor Trump/Harris to test the differences? (I could and should have made that comment after reading the first version, that's why I explicitly do not ask for revisions; this is just a thought that came to my mind when reading it once again).

typos and formal details: p. 4: "In line with this, is ..."  delete the comma

I do not believe that the decimals are necessary in means of word counts (e.g. p. 6 "M=173.79")

p. 9: "If case of"  "In case of"

Reviewer #2: I am very appreciative of the authors’ receptiveness to my constructive suggestions as well as their thoughtful efforts to address the reviewers’ major points in their cover letter and in this revision. The manuscript has improved in several respects, in particular a broader literature reviewer and a clearer connection between the analyses and the hypotheses they are trying to address.

My remaining issue is regarding the manuscript’s general discussion and conclusion section, which could benefit from more connection back to the literature raised in the introduction and theoretical background sections (there are virtually no citations in the discussion). It would be helpful for the reader to see a better integration between these sections and also help to ground some of the conclusions in previous research.

Relatedly, I would like the authors to more explicitly address limitations of their research as well as directions for future research that build on their findings. They give a couple of examples in specific studies (one limitation in the fifth study and one future research direction in the first study), but a broader perspective on these issues would help to address my previous review comment about missing some cohesion in bringing these measures together. I understand that no overarching theoretical framework exists, but the authors could find other ways to connect the different studies, such as a limitations/future research section. I also see ways that the data in different studies could be linked; for example, Trump naming himself among the top 5 named entries (Study 1, Figure 3) seems consistent with the data from the Group Appeal analysis where Trump used more individualistic language (Study 5). Having the authors reflect on findings across studies, wherever possible, would enhance the reader’s understanding of the findings and their implications.

7. PLOS authors have the option to publish the peer review history of their article (what does this mean?). If published, this will include your full peer review and any attached files.

Reviewer #1: No

Reviewer #2: No

---

## [Author Response · Author response to Decision Letter 2]

18 Apr 2025

please see the file "response to reviewers"

---

## [Decision Letter · Decision Letter 2]

30 Apr 2025

Red and blue language: Word choices in the Trump & Harris 2024 presidential debate

PONE-D-24-47133R2

Dear Dr. Bolognesi,

We’re pleased to inform you that your manuscript has been judged scientifically suitable for publication and will be formally accepted for publication once it meets all outstanding technical requirements.

Kind regards,

Michal Ptaszynski, PhD

Academic Editor

PLOS ONE

Additional Editor Comments (optional):

Reviewers' comments:

Reviewer's Responses to Questions

**Comments to the Author**

1. If the authors have adequately addressed your comments raised in a previous round of review and you feel that this manuscript is now acceptable for publication, you may indicate that here to bypass the “Comments to the Author” section, enter your conflict of interest statement in the “Confidential to Editor” section, and submit your "Accept" recommendation.

Reviewer #2: All comments have been addressed

2. Is the manuscript technically sound, and do the data support the conclusions?

Reviewer #2: Yes

3. Has the statistical analysis been performed appropriately and rigorously? 

Reviewer #2: Yes

4. Have the authors made all data underlying the findings in their manuscript fully available?

Reviewer #2: Yes

5. Is the manuscript presented in an intelligible fashion and written in standard English?

Reviewer #2: Yes

6. Review Comments to the Author

Reviewer #2: (No Response)

7. PLOS authors have the option to publish the peer review history of their article (what does this mean?). If published, this will include your full peer review and any attached files.

Reviewer #2: No

---

## [Editor Report · Acceptance letter]

PONE-D-24-47133R2

PLOS ONE

Dear Dr. Bolognesi,

I'm pleased to inform you that your manuscript has been deemed suitable for publication in PLOS ONE. Congratulations! Your manuscript is now being handed over to our production team.

Kind regards,

on behalf of

Dr. Michal Ptaszynski

Academic Editor

PLOS ONE